# A model for quick load analysis for monopile-type offshore wind turbine substructures

Signe Schløer[a], Laura Garcia Castillo[b], Morten Fejerskov[b], Emanuel Stroescu[b], and Henrik Bredmose[a]

[a]DTU Wind Energy, Nils Koppels Allé, Building. 403, DK-2800 Kgs. Lyngby, Denmark
[b]Universal Foundation, Langerak 17, DK-9220 Aalborg Øst, Denmark
*Correspondence to:* Signe Schløer (s.schloer@gmail.com)

**Abstract.** A model for Quick Load Analysis, QuLA, of an offshore wind turbine substructure is presented. The aerodynamic rotor loads and damping are precomputed for a land-based configuration. The dynamic structural response is represented by the first global fore-aft mode only and is computed in the frequency domain with phases using the equation of motion. The model is compared to the state of the art aeroelastic code Flex5. Both life time fatigue and extreme loads are considered in the comparison. In general there is good agreement between the two models. Some deviations for the sectional forces are explained in terms of the model simplifications. The differences in the sectional moments are found to be within 10 % for the fatigue load case and 10% for the extreme load condition.

## 1 Introduction

In order to ensure cost-efficient offshore wind farms, it is necessary to optimize the design. Particularly the substructures are expensive and can, according to Offshore Wind Project Cost OutLook (2014), account for 20 % of the total cost of energy.

It is often different companies who design the substructure and the wind turbine of an offshore wind turbine. The iteration process where the design suppliers of the wind turbine and the substructure send design loads back and forth slows the design process down.

The process is already time-consuming since extensive load-case simulations have to be made where different wind speeds and wave climates are combined. If instead a fully integrated simulation of the foundation and wind turbine is used, the design process will be faster and the number of uncertainties in the design will be reduced. However this approach is not always possible because the wind turbine manufacturers often do not want to share information about their wind turbines. Instead, in the preliminary design phase, the integrated simulation and optimization can be achived and accelerated with a simplified description of the loading from wind and a simple but fast dynamic model. This allows for optimization of the foundation in an early stage of the design.

Recently Schafhirt et al. (2015) combined a sub structuring technique, which is based on the principle of superposition of impulse responses, with the power of modern general purpose graphics processing units to compute the response of an offshore wind turbine subject to rotor loads. This method to perform simplified analysis of offshore wind turbines was found to the same

accuracy as standard aeroelastic models for distinct output locations as the overturning moment in the bottom of the tower but is 40 times faster for the case where only rotor loading on the substructure is considered.

Van der Tempel et al. (2005) presented a simple approach on how to speed up the fatigue load calculations by dividing the offshore wind turbine in to a turbine clamped at hub height with no support structure dynamics and a support structure. The analysis was linearized and made in the frequency domain by use of transfer functions, which according to Van der Tempel et al. (2005) is how the offshore oil and gas industry usually calculates the fatigue loads. The fatigue damage compared well to fatigue damage calculated in the time domain in the aeroelastic tool Bladed with a difference of approximately 8 %.

Smilden et al. (2016), also presented a simple model, but where the focus was to improve the performance of the control system. The model therefore also includes a wind model, drive-train and a controller possessing the main features of the wind turbine control system besides the mode shape based structural model. Smilden et al. (2016) included two tower mode shapes in the model.

In the present paper a model for **Qu**ick **L**oad **A**nalysis, QuLA, is presented. This is a fast model for calculation of dynamic loads of an offshore wind turbine tower and foundation. The wind loads are applied in a similar manner as Van der Tempel et al. (2005) while the structural model is based on a single mode shape. Compared to the above models, the wave kinematics are described in more detail without linearization and including wave nonlinearity for extreme load cases. The model is therefore suitable both for fatigue and the ultimate limit state. Compared to Van der Tempel et al. (2005) the aerodynamic damping is included as a function of mean wind speed, instead of being independent of the wind speed.

In the present paper the foundation is bottom fixed, however QuLA has been applied to a floating wind turbine too, see Lemmer et al. (2016) for preliminary results. The 10MW DTU reference wind turbine (Bak et al., 2013) is considered and the foundation is the Mono Bucket foundation of Universal Foundation[1]. The Mono Bucket consists of a shaft and a bucket as shown in figure 1. Compared to a monopile, the Mono Bucket has the advantage of very small noise impact during installation, reduced scour protection, and no need for a transition piece. So far a Vestas V90-3.0 MW offshore wind turbine has been erected on a Mono Bucket foundation in November 2002 in Frederikshavn harbour, Denmark. Besides, a met mast foundation for the Horns Rev 2 site was installed in March 2009 and decommissioned successfully in 2015,. Two other met mast foundations were installed at Forewind's Dogger Bank offshore wind site in September 2013 and were fully decommissioned in 2017, similarly to Horns Rev 2. In all three cases, the decommission was made by reversing the noise-free installation process. In order to make the Mono Bucket foundation commercial an industrialization and production evolution is needed. A fast numerical model to calculate the dynamic loads of the foundation is one of the tools applied in that process.

This paper investigates how well QuLA performs by comparing the model against the aeroelastic code Flex5, (Øye, 1996). The paper opens with a presentation of QuLA. Further, two different methods to include the aerodynamic damping are discussed. Hereafter the metocean data and the three load cases considered in the analysis is presented. Finally, the sectional inline force and overturning moment in different sections in the Mono Bucket and tower are considered for the load cases and both life time fatigue and extreme loads are analysed. The largest difference of 30% is found for the sectional inline force

---
[1]http://universal-foundation.com/

in the bottom of the Mono Bucket foundation, while the overturning moments compare well in most parts of the tower and Mono Bucket foundation with the largest difference being 10% . The design of the Mono Bucket foundation is confidential. Therefore, in this paper the results of the sectional forces and moments and response spectra are presented in normalized form.

This paper is part of a special issue of papers in Wind Energy Science journal and is an extended version of a previously
5   published paper, (Schløer et al., 2016a), published with IOP (from The Science of Making Torque From Wind conference). In the previous paper only one method to calculate the aerodynamic damping was considered and details on how the damping was calculated was left out. Furthermore, only two load cases were considered in the comparison of QuLA to Flex5 in the previously published paper. In the present paper the results are, compared to the previously published paper, improved mainly due to changes in the calculations of the aerodynamic damping and the addition of load case 1.3 to extend the analysis.

## Nomenclature

| | | | | |
|---|---|---|---|---|
| $\hat{a}$ | Peak in decay tests | | $t$ | Time, [s] |
| $A$ | Cross sectional area of pile, [m$^2$] | | $T_p$ | Peak wave period, [s] |
| $C_D$ | Hydrodynamic drag coefficient | | $u$ | Horizontal particle velocity, [m/s] |
| $C_{Da}$ | Aerodynamic drag coefficient | | $\dot{u}$ | Horizontal particle acceleration, [m/s$^2$] |
| $C_m$ | Hydrodynamic added mass coefficient | | u | Deflection, [m] |
| $C_M$ | Hydrodynamic inertia coefficient | | $w$ | Vertical particle velocity, [m/s] |
| $D$ | Pile diameter, [m] | | $W$ | Turbulent wind speed, [m/s] |
| $D_{aero}$ | Aerodynamic damping force, [kg/s] | | $x$ | Horizontal coordinate, [m] |
| $E$ | Modulus of elasticity, [ N/m$^2$] | | $z_N$ | Center of mass in the nacelle, [m] |
| $f$ | Frequency, [Hz] | | $z_{TT}$ | Tower top, [m] |
| $f_1$ | First natural frequency, [Hz] | | $z$ | Vertical coordinate, [m] |
| $f_p$ | Peak frequency, [Hz] | | $\lambda$ | Shear exponent for power low, eq. (1) |
| $f_{wave}$ | Hydrodynamic distributed force, [N/m] | | $\alpha$ | Generalized coordinate |
| $f_{Rainey}$ | Distributed Rainey force, [N/m] | | $\delta$ | Logarithmic decrement |
| $F_{aero}$ | Precalculated rotor force , [N] | | $\rho$ | Density of water , [kg/m$^3$] |
| $F$ | Force, [N] | | $\rho_a$ | Density of air, [kg/m$^3$] |
| $F_s$ | Rainay point force, [N] | | $\omega$ | Angular frequency , [rad/s] |
| $g$ | Gravity, [m/s$^2$] | | $\omega_0$ | Natural angular frequency , [rad/s] |
| $G_D$ | Generalized damping, [kg/s] | | $\sigma$ | Standard deviation |
| $G_F$ | Generalized force, [N] | | $\varphi$ | Shape function |
| $G_K$ | Generalized stiffness, [kg/s$^2$] | | $\zeta$ | Damping ratio |
| $G_M$ | Generalized mass, [kg] | | | |
| $H_s$ | Significant wave height, [m] | | | |
| $i$ | Imaginary unis | | | |
| $I$ | Area moment of inertia, [m$^4$] | | | |
| $I_T$ | Mass moment of inertia, [kgm$^2$] | | | |
| $\mathbf{K}$ | Stiffness matrix | | | |
| $KC$ | Keulegan-Carpenter number | | | |
| $\mathbf{K}_s$ | Translational and rotational soil spring matrix | | | |
| $L$ | Wave length, [m] | | | |
| $L_{eq}$ | Equivalent load, [N] or [Nm] | | | |
| $m$ | Distributed mass [kg/m] | | | |
| m | Wöhler exponent | | | |
| $\mathbf{M}$ | Mass matrix | | | |
| $M$ | Moment, [Nm] | | | |
| $M_{aero}$ | Precalculated rotor moment, [Nm] | | | |
| $M_{top}$ | Top mass, [kg] | | | |
| $N_{eq}$ | No of cycles | | | |
| $N_{s,i}$ | No of occurences of each stress range | | | |
| $P$ | Probability of occurrence | | | |
| $P_{rel}$ | Relative probability of occurrence | | | |
| $Re$ | Reynolds number | | | |
| $SWL$ | Still water level, [m] | | | |
| $S_i$ | Stress range, [N] or [Nm] | | | |

## 2 The numerical model QuLA

In QuLA, only the Mono Bucket foundation and wind turbine tower are considered and described as a simple Euler beam. On top of the beam a top mass, $M_{top}$, representing the rotor and nacelle is added. The top mass is placed in same height as the center of mass in the nacelle, $z_N$, 2.75 m above the tower top, $z_{TT}$, as illustrated in figure 2. The foundation is only considered down to the sea bed and the stiffness of the soil and lid and skirt of the bucket is described by a coupled translational and rotational spring, $\mathbf{K}_s$. The dynamic structural response is represented by the first natural mode only and the equation of motion is solved in the frequency domain.

The philosophy behind the model is to pre-calculate the aerodynamic forces in an aeroelastic model with a stiff foundation and tower for all considered wind speeds. Also the aerodynamic damping is pre-calculated for all considered wind speeds. The aerodynamic forces and damping are subsequently reused several times in QuLA for different tower and substructure configurations. The pre-computed aerodynamic force and damping can be made for a land-based turbine configuration and can thus be established as part of the turbine specifications by the manufacturer independent of the choice of substructure.

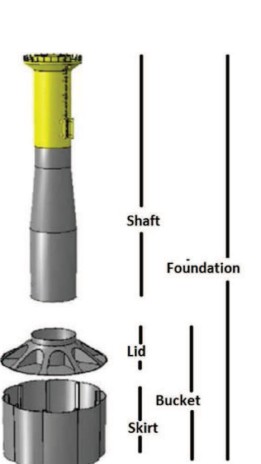

**Figure 1.** Mono Bucket foundation.

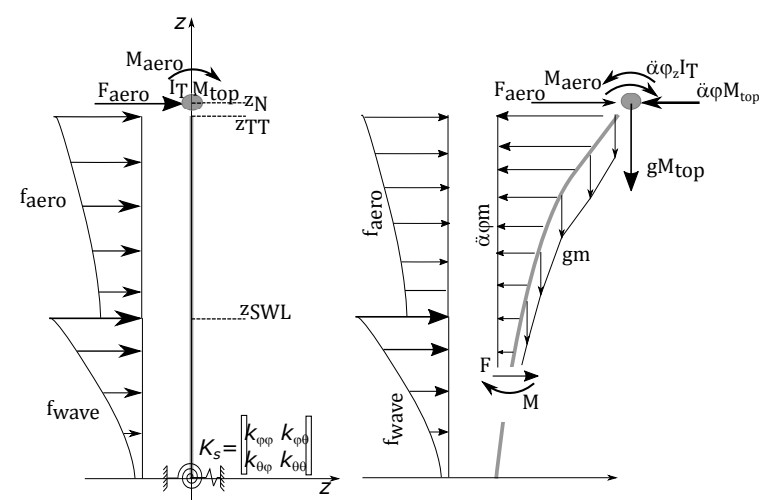

**Figure 2.** Left: Sketch of the beam and the external forces. Right: The external and internal forces which contribute to sectional force, $F$, and moment, $M$.

### 2.1 The external forces

The external forces are the distributed wave force and the turbulent wind force as seen in figure 2. The pre-calculated rotor shaft loads are applied as a time varying point force, $F_{aero}$ and overturning moment, $M_{aero}$ at center of mass in the nacelle

$z_N$. The force from the wind on the tower is also included and is calculated inside QuLA by the power law from IEC61400-3 (2009)

$$f_{aero}(x,t) = \frac{1}{2}\rho_a C_{Da} D \left( \left( \frac{x}{x_n} \right)^\lambda W(t) \right)^2,$$ (1)

Usually the shear exponent is designated as $\alpha$, but since $\alpha$ in this paper represent the generalized coordinate, the shear exponent
is instead designated as $\lambda$ with $\lambda = 0.14$ in load case 1.2 and load case 1.3 and $\lambda = 0.11$ in load case 6.1. Here $\rho_a = 1.225$ kg/m$^3$ is the density of air, $C_{Da} = 0.6$ is the aerodynamic drag coefficient, $D(z)$ is the diameter of the tower and $W$ is the turbulent wind speed at the nacelle.

     The wave kinematics and hydrodynamic force are also calculated inside QuLA. To enable fast calculations of the structural response, no stretching of the wave kinematics is applied and the wave kinematics are therefore only defined up to still water
level, $SWL$.

     In situations where fatigue loads are considered, linear wave theory is often sufficient to describe the wave kinematics, (Schløer et al., 2016b). For a known spectral shape (in this case JONSWAP) an irregular wave realization is characterised by the significant wave height $H_s$ and the peak wave period $T_p$. The linear irregular wave kinematics and loads are calculated in the frequency domain and afterwards transformed to the time domain using inverse Fast Fourier transformation in order to
include the nonlinear terms in the hydrodynamic force. The distributed hydrodynamic load on the structure is calculated by Morison's equation

$$f_{wave}(z,t) = \rho C_m A \dot{u} + \rho A \dot{u} + \frac{1}{2}\rho C_D D u|u|.$$ (2)

Here $\rho = 1025$ kg/m$^3$ is the density of water, $A(x)$ is the cross sectional area of the pile and $D(z)$ is the diameter of the pile. The horizontal particle velocity and acceleration are denoted $u$ and $\dot{u}$. The coefficients, $C_D$ and $C_m$, are the drag and added
mass coefficients, with $C_M = 1 + C_m$ being the inertia coefficient. The coefficients are functions of the Keulegan-Carpenter number, $KC$, and Reynolds number, $Re$, and are calculated following the recommendations in DNV-OS-J101 (2010). For irregular wave realizations $KC$ and $Re$ can, according to Sumer and Fredsøe (2006), be calculated from the standard deviation of the horizontal velocity at still water level and the mean wave period.

     The hydrodynamic damping due to the structural motion is considered small and neglected. Therefore it is the absolute
velocities, which are considered in the drag force, third term in (2).

     The added mass coefficient, $C_m$, is corrected for diffraction effects by the theory of MacCamy-Fuchs (MacCamy and Fuchs, 1954), which is valid for linear waves. The correction is important for waves with $D/L > 0.2$, where $L$ is the wave length. In a water depth of 50m it corresponds to wave frequencies larger than approximately $f > 0.19$ Hz. To include the diffraction effect, the added mass force is calculated in the frequency domain and afterwards transformed to the time domain.

In order to simultaneously include both the effect of wave irregularity and wave nonlinearity in the structural analysis, IEC61400-3 (2009) suggests to embed a large nonlinear stream function wave in the linear irregular wave time series to represent extreme waves. This is done in situations where ultimate loads (ULS) are considered. Following the work of Rainey (1989) and Rainey (1995), the Lagragian particle acceleration $du/dt$ is applied for these cases in the Morison equation instead of the Eulerian acceleration $\partial u/\partial t$ and further extended by the axial divergence correction term

$$f_{Rainey}(z,t) = \rho A C_m w_z u, \tag{3}$$

which according to Manners and Rainey (1992), corrects for the assumption that the cylinder is slender in the vertical direction. Here the vertical particle velocity is denoted $w$ and index "$z$" means that the variable is differentiated with respect to $z$.

Finally a point force should according to Rainey (1995) be added at the intersection with the water level

$$F_s(t) = -\frac{1}{2}\rho A C_m \eta_x u^2. \tag{4}$$

Here $\eta_x$ is the slope of the free surface elevation and represents the change of the free surface elevation along the pile-diameter. This force can be seen as a slamming force.

The Rainey terms, (3) and (4), are nonlinear contributions to the Morison force and are therefore only added to the Morison equation (2) in situations where a nonlinear single wave event is embedded in the irregular linear wave realization in the ULS-analysis.

## 2.2 The structural model

The structural dynamic deflection of the Mono Bucket and tower, u, is represented by a shape function, $\varphi$ and a generalized coordinate $\alpha$ as u $= \alpha(t)\varphi(z)$. Shape functions are often introduced when the equation of motion of a system is solved to decrease the number of degrees of freedom in the system and thereby the computational time. Only one shape function in the fore-aft direction is considered in QuLA. While this may not provide an accurate representation of the full deformation, it is here used for the purpose of approximating the associated inertia loads for the sectional forces, see (13)-(14). The shape function and the natural angular frequency, $\omega_0$ are found by considering a standard eigenvalue problem,

$$\mathbf{M}\ddot{\alpha}\underline{\varphi} + \mathbf{K}\alpha\underline{\varphi} = 0, \quad \text{where} \quad \alpha = \exp(i\omega_0 t) \Leftrightarrow \tag{5a}$$

$$-\mathbf{M}\omega_0^2\underline{\varphi} + \mathbf{K}\underline{\varphi} = 0 \Rightarrow \omega_0^2\underline{\varphi} = \mathbf{M}^{-1}\mathbf{K}\underline{\varphi}. \tag{5b}$$

The stiffness, $\mathbf{K}$, and mass matrix, $\mathbf{M}$, are calculated by the finite element method. Stiffness elements representing the stiffness from the soil-structure interaction, $K_s$ in figure 2, is calculated in the geotechnical software tool Plaxis (Brinkgreve et al., 2016) and is added to the stiffness matrix in the bottom of the pile. The top mass and mass moment of inertia around the

nacelle ($y$-axis), $I_T$, are added to the mass matrix in the top of the pile. To get the correct first natural frequency it is important to define $M_{top}$ and $I_T$ in same height as the center of mass in the nacelle, $z_N$.

The structural dynamics are calculated by the equation of motion

$$\ddot{\alpha}G_M + \alpha G_K + \dot{\alpha}G_D = G_F. \tag{6}$$

5     In order for the model to be fast, the equation of motion is solved in the frequency domain. In frequency domain the generalized coordinate can be expressed as

$$\alpha = \sum_{j=1}^{N_f} \hat{\alpha}_j \exp(i\omega_j t) + c.c., \tag{7}$$

where $\hat{\alpha}_j$ is a complex number, $\omega_j$ is the smallest angular frequency in the time series and $c.c.$ is the complex conjugate. The phase information of $\alpha$ is retained in (7).

10     The equation of motion

$$-\omega^2 G_M \hat{\alpha} + i\omega G_D \hat{\alpha} + G_K \hat{\alpha} = \hat{G_F} \Leftrightarrow \hat{\alpha} = \frac{\hat{G_F}}{-\omega^2 G_M + i\omega G_D + G_K} \tag{8}$$

then solves the linear response in frequency domain and can readily be transposed to the time domain by inverse FFT. By solving the equation in frequency domain the solution $\hat{\alpha}$ can then be solved at once for all time steps.

The generalized mass, $G_M$, and stiffness, $G_K$, can be obtained from (5a) by left-multiplication of $\varphi^\mathrm{T}$ and are given as

$$G_M = \int_{z=0}^{z_{TT}} m\varphi(z)^2 \, dz + M_{top}\varphi(z_N)^2 + I_T\varphi_z(z_N)^2, \tag{9}$$

$$G_K = \int_{z=0}^{z_{TT}} EI\varphi_{zz}(z)^2 \, dz. \tag{10}$$

Here $m(z)$ is the distributed mass of the tower and Mono Bucket foundation, $\varphi_z$ is the angular deflection of the shape function and $\varphi_{zz}$ is the curvature of the shape function. The stiffness factor is given by the modulus of elasticity, $E$, and the cross sectional area moment of inertia $I$. Further, the generalized damping, $G_D$, and generalized force, $G_F$, are given as

$$G_D = \zeta\frac{2G_K}{\omega_0} + D_{aero}, \tag{11}$$

$$G_F = \int_{z=0}^{z_{SWL}} \varphi f_{wave} \, dz + F_s + F_{aero}\varphi(z_{TT}) + M_{aero}\varphi_z(z_{TT}) + \int_{z_{SWL}}^{z_{TT}} \varphi f_{aero} \, dz. \tag{12}$$

Here, $\zeta$, is the damping ratio representing structural damping, soil damping and hydrodynamic radiation damping , here implemented as stiffness proportional damping. Further, $D_{aero}$ is the aerodynamic damping coefficient.

After the equation of motion is solved, the sectional forces and moments can be calculated. The external and internal forces, which contribute to the sectional forces and moments are shown in figure 2 and the forces and moments are calculated by integration over the structure above the point of interest as;

$$F(z^*,t) = -\ddot{\alpha} \int_{z^*}^{z_{TT}} m\varphi(z)dz - \ddot{\alpha}M_{top}\varphi(z_N) + \int_{z^*}^{z_{SWL}} f_{wave}dz + F_s + F_{aero}$$

$$+ \int_{z^*}^{z_{TT}} f_{aero}dz + \alpha g M_{top}\varphi_z(z_N) + \alpha g \int_{z^*}^{z_{TT}} m\varphi_z(z)dz \tag{13}$$

$$M(z^*,t) = -\ddot{\alpha} \int_{z^*}^{z_{TT}} m\varphi(z)[z - z^*]dz - \ddot{\alpha}M_{top}\varphi(z_N)[z_n - z^*] - \ddot{\alpha}I_T\varphi_z(z_N)$$

$$+ \int_{z^*}^{z_{SWL}} f_{wave}[z - z^*]dz + F_s[z_{SWL} - z^*] + M_{aero} + F_{aero}[z_{TT} - z^*]$$

$$+ \int_{z^*}^{z_{TT}} f_{aero}[z - z^*]dz + \alpha M_{top}g[\varphi(z_N) - \varphi(z^*)] + \alpha g \int_{z^*}^{z_{TT}} m[\varphi(z_{TT}) - \varphi(z)]dz, \tag{14}$$

where $g$ is the gravity. The first two terms in both equations are the contribution from the dynamics of the structure. When the equation of motion is solved, the Mono Bucket and tower are treated as an Euler beam, where the deflections are assumed small and only lateral loads are considered. Second-order contributions from the bending of the beam are therefore neglected in the solution in order for the model to be fast. However, in the sectional forces and moment the contribution from gravity due to the bending of the beam is included as stated in the last two terms in both equations. While this approach thus represent a difference in the forces applied for dynamics and sectional loads, it was found to improve the sectional loads in the comparison to Flex5.

## 2.3 Shape function and eigenfrequency

The complete shape function of both the tower and bucket foundation in Flex5 is compared to the shape function of QuLA in figure 3. The shape functions are close to being identical. The deviation between the first natural frequency of the two models is 1%. The difference is caused by differences in the models: In Flex5 the gravity's contribution to the bending of the pile is included in the equation of motion, which gives a larger moment of inertia and therefore a lower frequency. In QuLA the contribution of the gravity of both the top mass, representing the blades, hub and nacelle and the tower is only included in the sectional forces calculated after the equation of motion is solved.

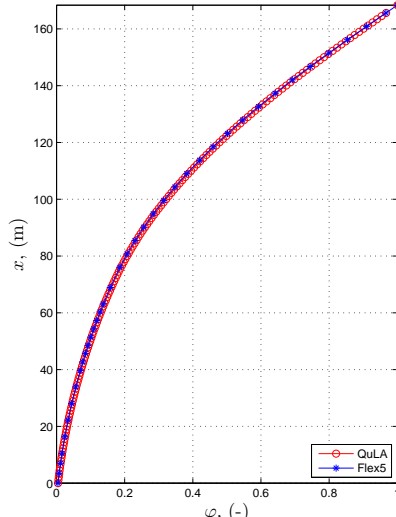

**Figure 3.** The shape function.

## 2.4 The aerodynamic damping

As the structural dynamics is included in QuLA, it is also necessary to include the aerodynamic damping. If the structural motion is in same direction as the wind velocity, the relative velocity which the aerodynamic forces are a function of, decreases and thereby also the forces. Since the aerodynamic forces are included as point forces in QuLA and since the equation of motion is solved in the frequency domain, the aerodynamic damping can only be added as a viscous linear damping force, where the damping coefficient is a function of the mean wind speed.

Two different methods to calculate the damping coefficients are presented below and compared for load case 1.2 in section 4.1.

### 2.4.1 Standard deviation of pile displacement

In this approach the target is to have the same standard deviation of the pile displacement at the top of the tower. Therefore the tower top displacement has to be calculated in advance in Flex5 or another aeroelastic model for all considered cases. In QuLA, when the equation of motion is solved, a loop is included, where the aerodynamic damping is increased until the standard deviation is the same for Flex5 and QuLA. The standard deviation is calculated as

$$\sigma = \sqrt{\overline{\left(\mathrm{u}_{NN} - \overline{\mathrm{u}_{NN}}\right)^2}}, \tag{15}$$

where $\mathrm{u}_{NN}$ is the tower top displacement.

In figure 4-5 two examples of the tower top displacements calculated in Flex5 and QuLA for $W = 4.16$m/s and $W = 14.55$m/s are shown for load case 1.2. For the small wind speed the two models compare very well, however as the wind increases differences between the two models are more visible. This is due to differences in how the models are solved. In Flex5 the aerodynamic damping is a function of time, while in QuLA it is represented by a constant linear damping coefficient for each mean wind speed. Furthermore, in QuLA only one degree of freedom is used and the gravity's contribution to the deflection is not included in QuLA as mentioned in section 2.3.

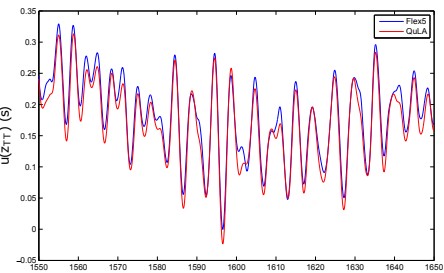

**Figure 4.** Tower top displacement for $W = 4.16$ m/s. The aerodynamic damping in QuLA is based on standard deviation of pile displacement.

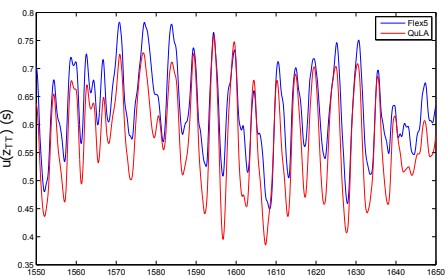

**Figure 5.** Tower top displacement for $W = 14.55$ m/s. The aerodynamic damping in QuLA is based on standard deviation of pile displacement.

### 2.4.2  Decay tests

The aerodynamic damping force, $D_{aero}$, is calculated alternatively in Flex5 by decay tests as function of wind speed. To calculate the damping force both turbulent and steady wind speeds are considered. For both cases two simulations are run. One where a starting velocity of the wind turbine tower and foundation is applied and one simulation without a starting velocity. The logarithmic decrement is calculated from the difference between the two simulations with and without a starting velocity.

All degrees of freedom are active, however the rotor speed is kept constant and the pitch angle and rotational speed are given initial values in accordance with the wind speeds considered. According to Salzmann and Van der Tempel (2005) this method works well for constant speed wind turbines and compares well with other simple methods as the Garrad method Freris and Freris (1990), Kühn's closed-form model Kühn (2001) or van der Tempel's method, Van Der Tempel (2006). However, for a pitch regulated wind turbine with varying rotor speed, which is the case for the DTU 10MW wind turbine, such simple methods may provide, according to Salzmann and Van der Tempel (2005), a less accurate estimation above rated wind speed, where the pitch regulation begins. However, the logarithmic decrement in Qula can only be represented by a single value as function of the mean wind speed. Therefore, the logarithmic decrement above rated wind speed is still found by keeping the pitch and rotor speed constant.

The logarithmic decrement is calculated as

$$\delta = \frac{1}{j} \log\left(\frac{\hat{a}_1}{\hat{a}_j}\right), \quad \text{where} \quad j = 2, 3....,$$ (16)

where $\hat{a}_1$ is the first peak considered in the time series and $\hat{a}_j$ is the $j$'th amplitude following $\hat{a}_1$. The relation between the logarithmic decrement, damping ratio, $\zeta$ and the damping force, $D_{aero}$, which is used in the dynamic analysis, is

$$\delta = \frac{2\pi\zeta}{\sqrt{1-\zeta^2}},$$ (17)

$$D_{aero} = \zeta 2\sqrt{G_M G_K},$$ (18)

where $G_M$ and $G_K$ are the generalised stiffness and mass, cf. section 2.2.

In figure 6-7 the decay tests for a steady and turbulent wind speed of 14 m/s are shown. In the top figures the displacements in the top of the tower are shown both for the case where the tower has an initial velocity of $U_{init} \sim 1.1$m/s and the one without

10   an initial velocity and in the bottom the subtracted displacements are shown. The logarithmic decrement has been estimated for the four peaks, both positive and negative, following the largest peak and then averaged,

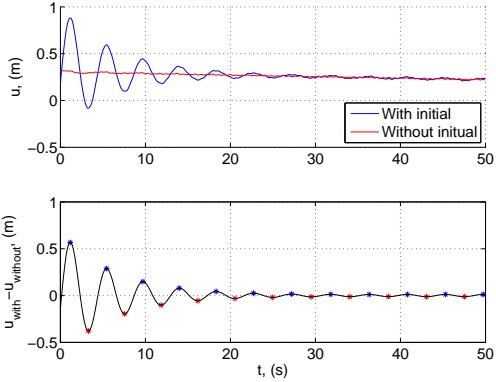

**Figure 6.** Decay test for a steady wind of 14 m/s.          **Figure 7.** Decay test for a turbulent wind of 14 m/s.

In figure 8 the damping ratios as function of both steady and turbulent wind speed are shown for three initial tower velocities. In figure 8 the average of the six curves is also shown. It can be observed that the damping ratio is very similar across the initial tower velocities.

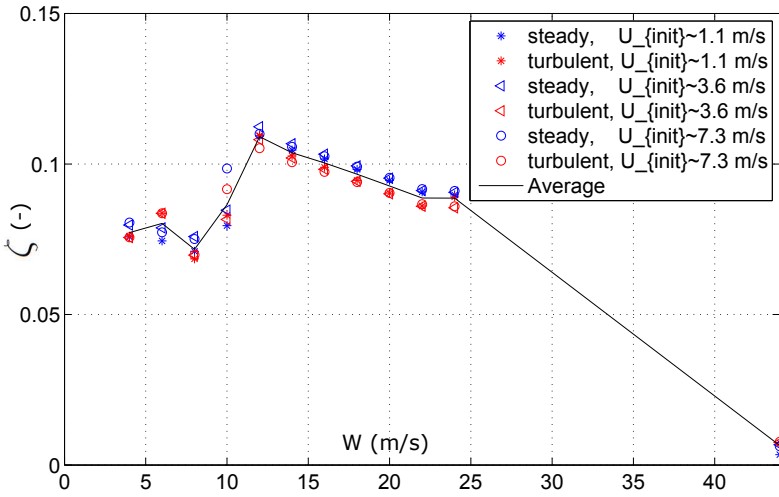

**Figure 8.** The damping ratio as function of wind speed for different decay-tests.

### 2.4.3 Comparison of the damping ratios

In figure 9 the damping ratio as function of the wind speed from cut-in to cut-out wind speed is shown for the two different methods used to calculate the aerodynamic damping force. The trend is similar for both damping curves. The damping ratio is constant for small wind speeds with a value between 7-8 % for the decay tests and 2 % for the standard deviation of tower top

displacement but starts to increase before rated wind speed.

For the decay tests the largest damping ratio of 10.5 % is reached for a wind speed of 12 m/s. Above rated wind speed, the damping ratio decreases and is approximately 9% for a wind speed of 25 m/s. For wind speeds between 10 and 17 m/s, the damping ratio based on the standard deviation increases from 2- to 10.3% , where after it decreases and is 7% for a wind speed of 25 m/s.

The damping ratio, based on decay tests, is larger than the one based on the standard deviation of the tower displacement except for $W \sim 17$ m/s, where the damping based on the tower top displacement is largest. This might be because the standard deviation puts more weight to the low-amplitude motion. In the decay tests, the damping seems to become smaller for low amplitude motion, see figure 7 lower plot, for $t > 30$ s. The reason for the large increase in the damping ratio based on the standard deviation above rated wind speed $W = 17$ m/s can be because this method assumes that the tower deflection is the

same for Flex5 and QuLA. This is not correct as explained in section 2.3. As the wind speed increases the tower has a larger deflection and the contribution from the gravity is therefore larger. This contribution is not included when QuLA calculates the deflection.

Both methods require some preliminary work to calculate the viscous damping force, to be used in QuLA. Either decay tests have to be made or the displacement in the top of the tower has to be calculated in an aeroelastic tool. However, as the

foundation is very stiff, it is not believed that the foundation contributes significantly to the damping force. Therefore, the

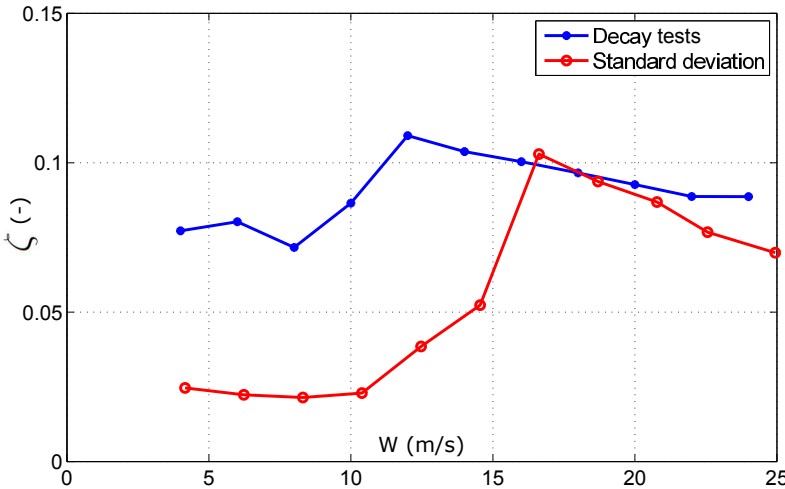

**Figure 9.** The damping ratio as function of wind speed for the two methods to calculate the damping.

preliminary work can be made for a land based wind turbine, and the aerodynamic damping reused several times as long as the wind turbine and tower is not changed.

How the different damping curves influence the performance of QuLA is investigated in section 4.1.

## 3 Metocean data and structure

The load cases in the present analysis are based on the metocean data from the artificial site "'K13 Deepwater Site"' from the Upwind-project (Fischer et al., 2010). The water depth is $h = 50\,\mathrm{m}$. Three load cases are studied, load case 1.2 which consider the fatigue limit state (FLS) and load case 1.3 and 6.1 which consider the ultimate limit state (ULS). The time series of each wind and sea state is 1 hour long which corresponds to six seeds of 600 s. In load case 1.2 the wind turbine operates, and the wind speed ranges from 4m/s to 25 m/s with an interval of 2m/s using a normal turbulence model. The already lumped sea

states presented in Fischer et al. (2010) are used together with the wind speeds. Since fatigue loads are considered, the wind speeds probability of occurrence is taken into account.

The wind speeds and the corresponding probability of occurrence, $P_r$, turbulence intensity, $I$, and sea states are stated in table 1.

In load case 1.3 the wind turbine operates. operates. The wind speeds are the same as for load case 1.2, but the turbulence

intensity is now based on an extreme turbulence model. The significant wave height is the expected wave height conditioned on the wind speed

$$H_s = E[H_S|W] = \sum_i H_{s,i} P_{rel}, \tag{19}$$

where $P_{rel}$ is the relative probability of occurrence of each significant wave height conditioned on the considered wind speed. The range of peak wave periods appropriate to each $H_s$ should be taken into account and the one resulting in the largest load should be used in the ULS-analysis. Further, if the peak wave period corresponding to the first natural frequency, $f_1 = 1/Tp$ is inside the considered range this wave period should also be considered. The same applies to higher harmonics of the wave peak

period, i.e. multiples of the peak wave frequency, $2f_p$ and $3f_p$, as this will cause a a larger excitation of the structure. In the present analysis, the largest and smallest wave peak period which occur, are considered. The wind speed, turbulence intensity and corresponding $H_s$ and $T_p$ values are stated in table 2. Also the periods in between the smallest and largest $T_p$-value, for which the frequency or its multiples are equal to the first natural frequency are considered. However, due to confidential design, these frequencies are not written in the table, but a "+" indicates for which wind speeds they occur.

In load case 6.1 the wind turbine is parked and the wind speed is 44.03 m/s. The corresponding sea state has a significant wave height of $H_s = 9.40$m and a peak period of $T_p = 10.87$s.

    In ULS situations an irregular linear wave time series of 1 hour length plus 100 s of transient time is first created. For every 600 s the largest wave in the interval is replaced with a nonlinear regular stream function wave with a wave height of $H = 1.86H_s$, IEC61400-3 (2009). The corresponding wave period should according to IEC61400-3 (2009), be chosen as the

period in the interval

$$11.1\sqrt{H_s/g} < T < 14.3\sqrt{H_s/g}, \tag{20}$$

which results in the largest load. In the present analysis six wave periods from $11.1\sqrt{Hs/g}$ to $14.3\sqrt{Hs/g}$ were considered for load case 6.1. It was found that for the present structure the largest load occurred for the smallest wave period, $T = 11.1\sqrt{H_s/g} = 10.87$s in load case 6.1. In load case 1.3 the same ratio, $T = 11.1\sqrt{H_s/g}$ is also used.

| $W$ | $P_{rel}$ | $I$ | $H_S$ | $T_p$ |
|---|---|---|---|---|
| (m/s) | (-) | (-) | (m) | (s) |
| 4.16 | 0.11 | 0.29 | 1.10 | 5.88 |
| 6.23 | 0.14 | 0.23 | 1.18 | 5.76 |
| 8.31 | 0.16 | 0.20 | 1.31 | 5.67 |
| 10.39 | 0.15 | 0.18 | 1.48 | 5.74 |
| 12.47 | 0.13 | 0.17 | 1.70 | 5.88 |
| 14.55 | 0.11 | 0.16 | 1.91 | 6.07 |
| 16.62 | 0.08 | 0.15 | 2.19 | 6.37 |
| 18.70 | 0.05 | 0.15 | 2.47 | 6.71 |
| 20.78 | 0.03 | 0.14 | 2.76 | 6.99 |
| 22.56 | 0.02 | 0.14 | 3.09 | 7.40 |
| 24.94 | 0.01 | 0.14 | 3.42 | 7.80 |

Table 1. The wind speeds and the corresponding probability of occurrence, turbulence intensity and sea states for load case 1.2, Fischer et al. (2010).

| $W$ | $I$ | $H_S$ | $T_{p,min}$ | $T_{p,max}$ | $T_p$ for $f_1 = 1/Tp$ | $T_p$ for $f_1 = 2/Tp$ |
|------|------|------|------|------|------|------|
| (m/s) | (-) | (-) | (m) | (s) | (s) | (s) |
| 4.16 | 0.82 | 5.88 | 4 | 11 | + | + |
| 6.23 | 0.90 | 5.76 | 4 | 11.5 | + | + |
| 8.31 | 1.05 | 5.67 | 4 | 11.5 | + | + |
| 10.39 | 1.23 | 5.74 | 4 | 11. | + | + |
| 12.47 | 1.46 | 5.88 | 5 | 9 | - | - |
| 14.55 | 1.72 | 6.07 | 5 | 8 | - | - |
| 16.62 | 2.07 | 6.37 | 5 | 9 | - | - |
| 18.70 | 2.38 | 6.71 | 5 | 10 | - | + |
| 20.78 | 2.80 | 6.90 | 5 | 8 | - | - |
| 22.56 | 3.13 | 7.40 | 7 | 9 | - | - |
| 24.94 | 3.58 | 7.80 | 7 | 10 | - | + |

**Table 2.** The wind speeds and the turbulence intensity and sea states for load case 1.3.

The wind turbine is the 10 MW DTU reference wind turbine, (Bak et al. (2013)). The wind turbine has a rated wind speed of 11.4 m/s and a rated rotor speed of 9.6 RPM. The rotor diameter is 178.3 m and hub height is 119 m. The rotor and nacelle mass are 229 tons and 446 tons, respectively.

The first natural frequency of the structure is in between the 1P and 3P frequency interval of the wind turbine (1P=0.115–0.159 Hz). In both Flex5 and QuLA a logarithmic damping decrement $\delta = 2\pi\zeta = 0.06$ is included to represent soil damping, structural damping of the Mono Bucket and tower and hydrodynamic radiation damping.

## 4   Results

In order for QuLA to be a useful tool in the design-process, the model has to be faster than a more advanced aeroelastic model. Before QuLA can be used it is necessary to precalculate the stochastic point loads, $F_{aero}$ and $M_{aero}$ and the aerodynamic damping. Though, once they are calculated they can be used repeatedly in the design process.

To calculate a single wind and sea state on a Microsoft Windows machine with a clock rate of 2.30 GHz QuLA is 40 times faster than Flex5, while on a Linux cluster machine with a clock rate of 1.9 GHz QuLA is 3.3 times faster. It is belived that this can be speeded up to similar performance as at the Windows machine. QuLA is further parallelised, and can on a HPC-cluster calculate in parallel all 11 wind and sea states of load case 1.2 in approximately 45s.

### 4.1   Fatigue limit state

Load case 1.2 considers the fatigue limit state during operation. For this load case the different methods to calculate the aerodynamic damping are compared.

In figure 10-11 the probability of exceedance, $P$, of the positive peaks in the 1 hour time series of the sectional forces and moments in five sections of the Mono Bucket and tower are shown for the case with $V = 10.39$ m/s, $H_s = 1.48$ m and

$T_p = 5.74$ s. Only the deflection in the fore-aft direction is calculated in QuLA and therefore only the forces and moments in the fore-aft direction are considered in the analysis. The forces and moments are normalized with the largest force and moment peak at the sea bed in the Flex5-calculation.

Considering the force peaks in the tower QuLA compares best to Flex5 when the damping is based on decay tests (figure 10), while in the Mono Bucket it is when the damping is based on standard deviation of the tower top displacements (figure 11). For both methods the difference between QuLA and Flex5 is largest in the Mono Bucket. Comparing the moments, which are usually more relevant for the design, the two models are very similar with largest difference at the sea bed. Considering the exceedance probability curves of the moment peaks, the differences in the damping are easy to identify. Compared to Flex5 the moment peaks of QuLA are smallest when the damping is based on decay tests, while the opposite is seen when the damping is based on standard deviation of the tower top displacement.

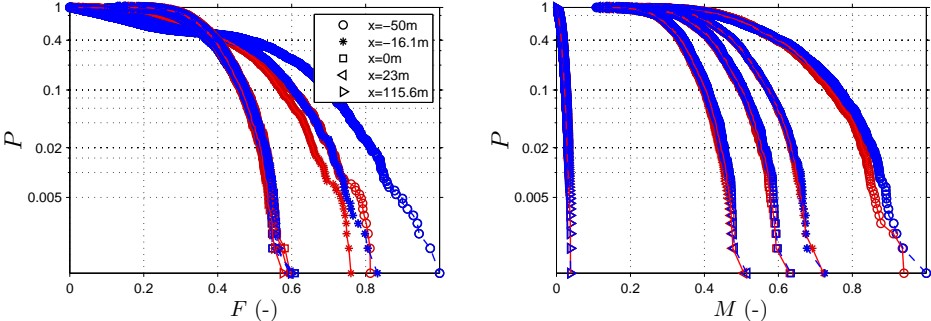

**Figure 10.** Probability of exceedance of the positive peaks in the time series of the sectional forces and moments for load case 1.2. Aerodynamic damping based on decay tests. The forces and moments are normalized with the largest force and moment peak at the sea bed in the Flex5-calculation. Blue: Flex5. Red: QuLA

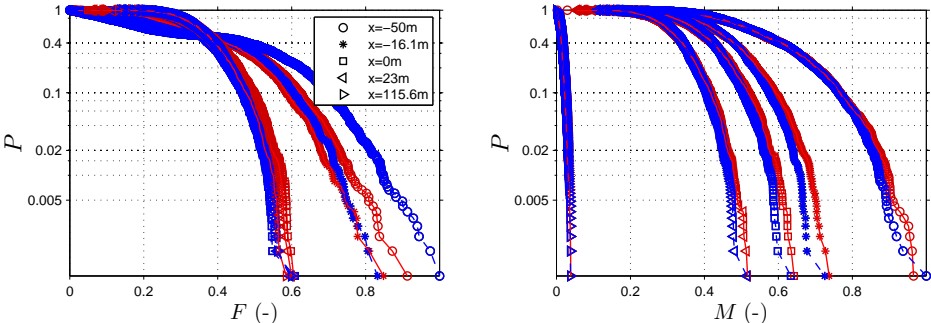

**Figure 11.** Probability of exceedance of the positive peaks in the time series of the sectional forces and moments for load case 1.2. Aerodynamic damping based on standard deviation of tower top displacement. The forces and moments are normalized with the largest force and moment peak at the sea bed in the Flex5-calculation. Blue: Flex5. Red: QuLA

In fatigue analysis, equivalent loads, $L_{eq}$, can be used as a reference loading and represents one load range value that for a certain number of cycles, $N_{eq} = 10 \cdot 10^6$, results in the same damage level as the history of investigated fatigue loads. It is calculated as

$$L_{eq} = \left( \sum_j \left( \sum_i \frac{N_{s,i} S_i^{\mathrm{m}}}{N_{eq}} \right) P_{rel,j} \right)^{\frac{1}{\mathrm{m}}}. \tag{21}$$

In (21) $N_{s,i}$ is the number of occurrences of each stress range, $S_i$, for the considered wind and sea state, $j$. The equivalent loads are calculated for the sectional forces and moments using a Wöhler exponent of $\mathrm{m} = 4$ and taking the wind and sea states probability of occurrence into account.

In figure 12-13 the ratio of the equivalent forces and moments of QuLA to those of Flex5 (QuLA/Flex5) throughout the tower and Mono Bucket are shown, both when the aerodynamic damping force is based on decay tests and standard deviation of tower top displacements.

The variation in the ratios in the tower and Mono Bucket is the same for the two damping methods, but it is clear that the damping based on the decay tests result in the best agreement with Flex5 results. Instead of having ratios around 1 in most part of the structure, which is seen when the damping is based on decay tests, the ratios is approximately 1.15, when the damping is based on tower top displacements. However, with both damping methods the difference between the equivalent forces of QuLA to those of FLex5 increases from the top of the monopile and down to the bottom. Near the sea bed the difference between the equivalent forces of QuLA to those of Flex5 is 0.7. The largest difference is here for the aerodynamic damping based on decay tests.

This change from the tower to the monopile can be explained by considering a sequence of the time series and response amplitude spectra of the 1 hour time series of the sectional forces at the intersection between the Mono Bucket foundation and tower (26 m above SWL) and at the sea bed as seen in figures 14-15. The aerodynamic damping force are based on decay tests. The forces are due to the wind and sea state with a wind speed of 10.39m/s, since this is found to contribute the most to the equivalent loads. The force time series are normalized with the largest force at the sea bed in the Flex5 calculations.

The energy around the first natural frequency is captured well by QuLA. Using Flex5 a big amount of energy is also found at the second natural frequency of the tower and Mono Bucket - in particular at the sea bed. Since QuLA only has one degree of freedom and thus only one natural mode, no energy is observed in QuLA at this frequency. The main part of the modal energy of the second natural frequency is distributed in the Mono Bucket, which explains why the difference between the two models at the second natural frequency is largest at the sea bed and why the ratio of the equivalent forces in figures 12-13 decreases throughout the Mono Bucket. This difference could be reduced by including a second degree of freedom in QuLA, as was done by Smilden et al. (2016). However, this will also double the complexity of the model, and focus has been to develop a very

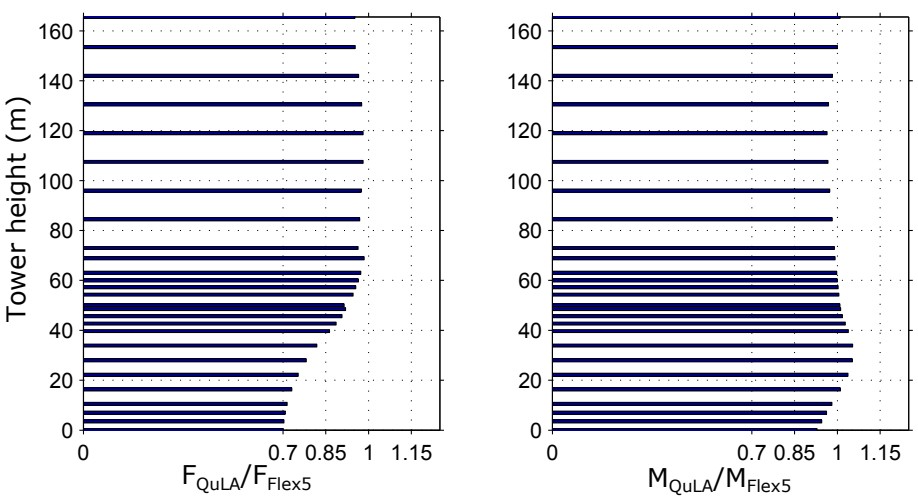

**Figure 12.** The ratio of the equivalent loads of QuLA to those of Flex5 for load case 1.2 in all sections in the tower and Mono Bucket. Aerodynamic damping based on decay tests.

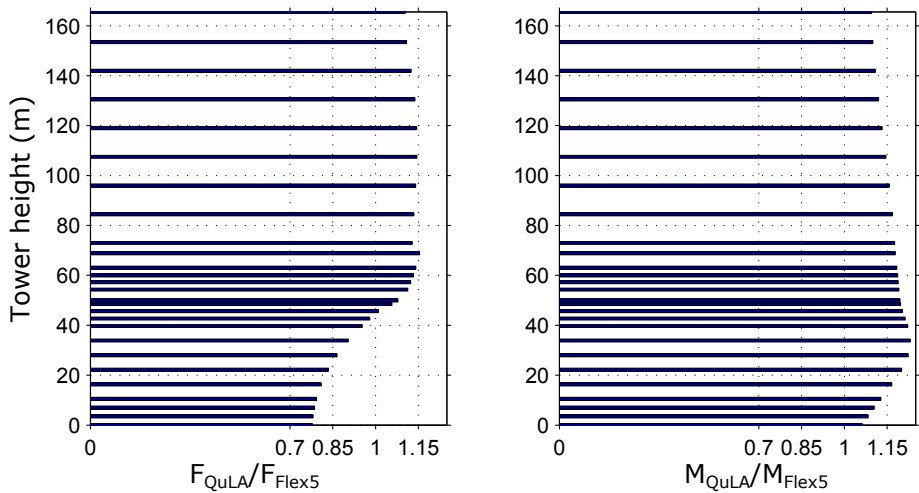

**Figure 13.** The ratio of the equivalent loads of QuLA to those of Flex5 for load case 1.2 in all sections in the tower and Mono Bucket. Aerodynamic damping based on standard deviation of tower top displacement.

simple and fast model.

The difference between the equivalent moments in the mono-bucket varies. Considering the ratios when the damping in QuLA is based on decay tests, the ratio changes from 1 to 1.05 from still water level to approximately 20 m above the sea bed, where after the equivalent loads of QuLA at the sea bed again becomes smallest with a ratio of 0.93 relative to those of Flex5. The same trend is seen when the damping is based on tower top displacements, but a factor of 0.15 approximately should be

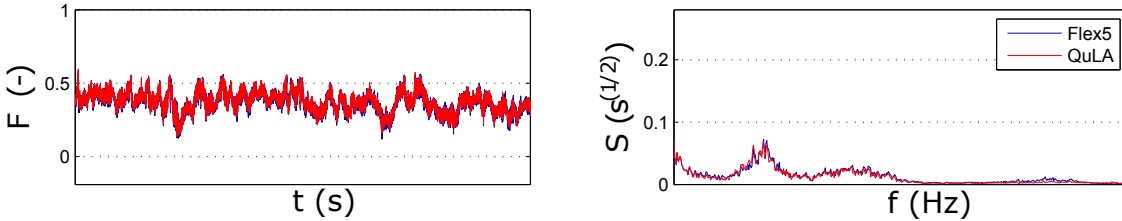

**Figure 14.** Sectional force 26 m above SWL for load case 1.2. The force time series are normalized with the largest force at the sea bed in the Flex5 calculations.

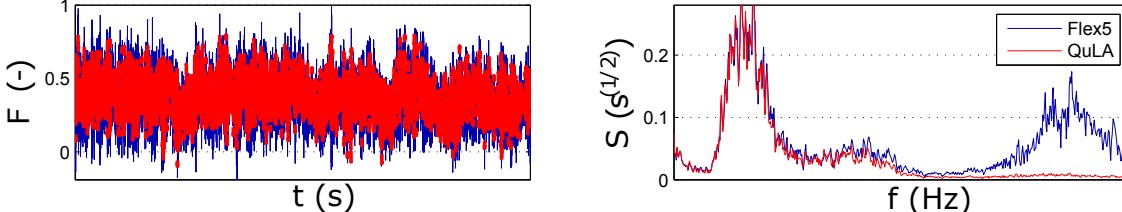

**Figure 15.** Sectional force at sea bed for load case 1.2. The force time series are normalized with the largest force at the sea bed in the Flex5 calculations.

added to the ratios. The reason that there is such a difference between the equivalent forces and moments is that the moments not only depend on the size of the overlying forces but also on the size of the moment arm.

The difference in the results obtained with the two methods to calculate the damping, must be because the decay tests only are based on Flex5 results, while the method with the tower top displacements are based on the assumption that the tower deflection is the same for Flex5 and QuLA, which is not correct for all wind speeds. In the ULS analysis in next section, the damping is therefore is based on the decay tests.

In the comparison of the equivalent loads of QuLA and Flex5, it is important to note that instead of the equivalent load ratio, the damage ratio could also be considered which differs from the equivalent load ratio by the power of the Wöhler exponent, $m$. Thus, the difference between the models is larger with that measure.

## 4.2 Ultimate limit state

Load case 1.3 and 6.1 consider the ultimate load state, ULS. To calculate the ultimate loads the 1 hour time series of the forces and moments for each wind and sea state in the load case are divided into 6x600s intervals. In each interval the largest load is found and the average of these six loads calculated. This approach is consistent with the IEC 61400-3 code, clause 7.5.1 for load case 1.3. For load case 6.1a, clause 7.5.1 states that six 1 hour realizations should be considered, unless it can be demonstrated that the extreme response is not affected by application of shorter realizations. Constrained wave methods is mentioned as one way of enabling shorter realizations. This approach has been adopted for the present study. For some realizations, we found that the largest loads occurred at events outside of the embedded constrained wave. This is further discussed in section 4.2.2.

### 4.2.1 Load case 1.3

In load case 1.3 the wind turbine operates.

In figure 16 the probability of exceedance, $P$, of the positive force peaks and moment peaks in the 1 hour time series with $W = 12.5$m/s, $H_s = 1.46$m and $T_p = 9$s in five sections of the Mono Bucket and tower are shown. In the tower the forces and
moments compare well. In the Monobucket a large difference between the forces is seen, which is due to the excitation of the second structural frequency in Flex5 as was seen for load case 1.2. The difference in the forces also influence the moments at the sea bed, where Flex5 has the largest moment. The moments are in much better agreement between the two models than the forces.

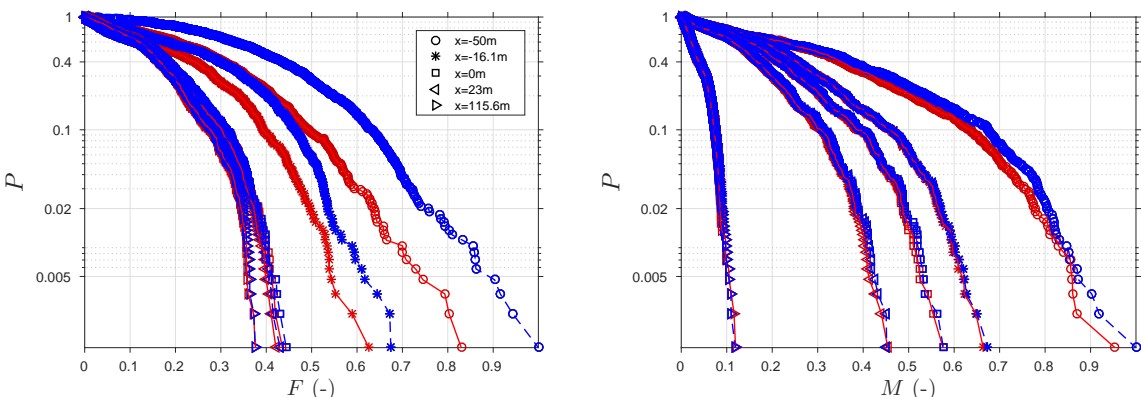

**Figure 16.** Probability of exceedance of the positive peaks in the time series of the sectional forces and moments for load case 1.3. The forces and moments are normalized with the largest force and moment peak at the sea bed in the Flex5-calculation. Blue: Flex5. Red: QuLA

Considering the ratios of the ultimate limit states of QuLA to those of Flex5 in figure 17, the same is seen. In the tower, the
differences between the ultimate sectional forces and moments of QuLA and Flex5 are not more than 2% and 4%, respectively. Flex5 has the largest ultimate moments in all sections, while the ultimate sectional forces, of Flex5 are largest in the top of the tower and the ultimate sectional forces of QuLA are largest in the bottom of the tower. In the Mono Bucket FLex5 has the largest ultimate forces due to the excitation of the second natural frequency. The difference between the models increases from the top to the bottom of the Mono Bucket and at the sea bed the ratio is 0.85. The effect of the second natural frequency is also
visible in the ultimate moments, but the effect is not as strong, since the forces in the tower still contribute more to the moment at the sea bed, where the ratio between the two models is 0.95. Usually the moment is more relevant for the design than the sectional force since it contributes more dominantly to the local stress.

### 4.2.2 Load case 6.1

Load case 6.1 considers a storm condition. The wind turbine is therefore idled, and the aerodynamic force and damping are
therefore small. The contribution from the wave force is thus expected to be significant.

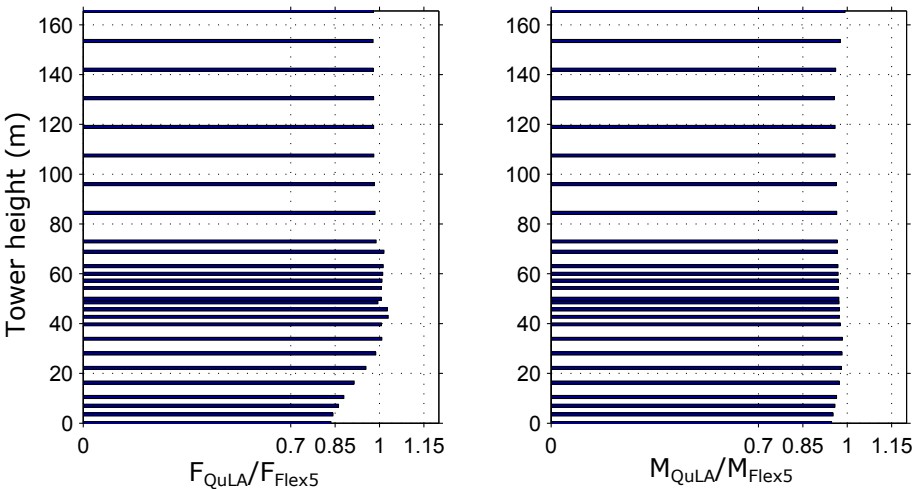

**Figure 17.** The ratio of the ultimate loads of QuLA to those of Flex5 for load case 1.3 in all sections in the tower and Mono Bucket.

In figure 18 the probability of exceedance, $P$, of the positive peaks in the 1 hour time series of the sectional forces and moments in five sections of the Mono Bucket and tower are seen. Note that the six embedded stream function waves are part of the exceedance probability curve. Usually, the response of these waves would form the basis of the ULS load value. In the following, however, we investigate the full response through the 1 hour duration of calculations to obtain insight into the model

5  performance.

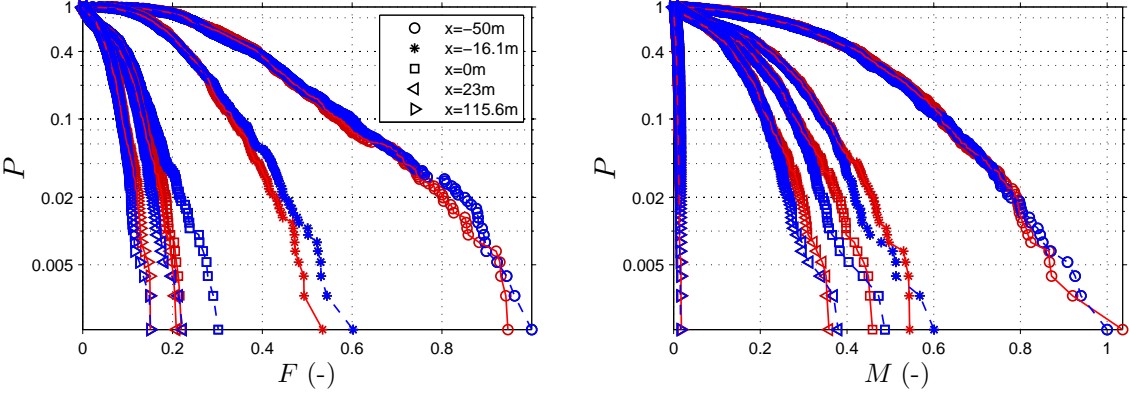

**Figure 18.** Probability of exceedance of the positive peaks in the time series of the sectional forces and moments for load case 6.1. The forces and moments are normalized with the largest force and moment peak at the sea bed in the Flex5-calculation. Blue: Flex5. Red: QuLA

QuLA has the largest force peaks for high probability of exceedance while Flex5 has the largest force peaks for low probability of exceedance, however in the tower, the probability of the force peaks are quite equal. At still water level there is a large difference between the curves of the two models, in particular for $P < 0.03$. The peaks of Flex5 are largest, which is caused

by the Wheeler stretching in Flex5, which stretches the wave kinematics up to the free surface elevation instead of only being defined to still water level as in QuLA. In the Mono Bucket 26 m above the sea bed, the difference between the two models is still significant but smaller. At the sea bed the force-curves of the two models are again quite equal.

The probability curves of the moments of the two models are more equal in all five sections. Particularly the largest moments,
which are important in ULS, compare well. To compare the dynamics of the two models a sequence of the time series and response amplitude spectra of the 1 hour time series of the sectional forces and moments at the intersection between the tower and Mono Bucket and at the sea bed are considered, figures 19-20. In the tower, the energy of the force and moment is located around the first natural frequency and it is clear that QuLA contains most energy at this frequency. This is opposite to the time series, which indicate that the response of Flex5 contains most energy. However, this does not account for the whole time
series, for which the spectra are based on.

At the sea bed, the energy is located both at the wave peak frequency and at the first natural frequency. The energy distribution of the force is very similar in the two models, while for the moments QuLA contains most energy. In the time series the forces of the two models are very similar when a stream function wave is embedded into the wave realization - indicated with an arrow in the figure. However, for the chosen sequence of the time series the moments are not largest when the stream function wave
is embedded. Instead the moments are largest in the beginning of the time sequence where the wave kinematics are described by linear wave theory. This means that for the stiffness and natural frequency of this foundation, the linear wave kinematics can also result in the largest moments. In other parts of time series, though, the embedded stream function wave results in the largest overturning moment at the sea bed. Still, this shows that the dynamic forces caused by the structural motion - and not only the static forces, are important in ULS. Furthermore, the presence of larger loads in some linear wave driven events
indicate that for a full design study longer realizations should be included following the IEC design code. For the present study, though, this has been omitted since it does not affect the comparison between QuLA and Flex5.

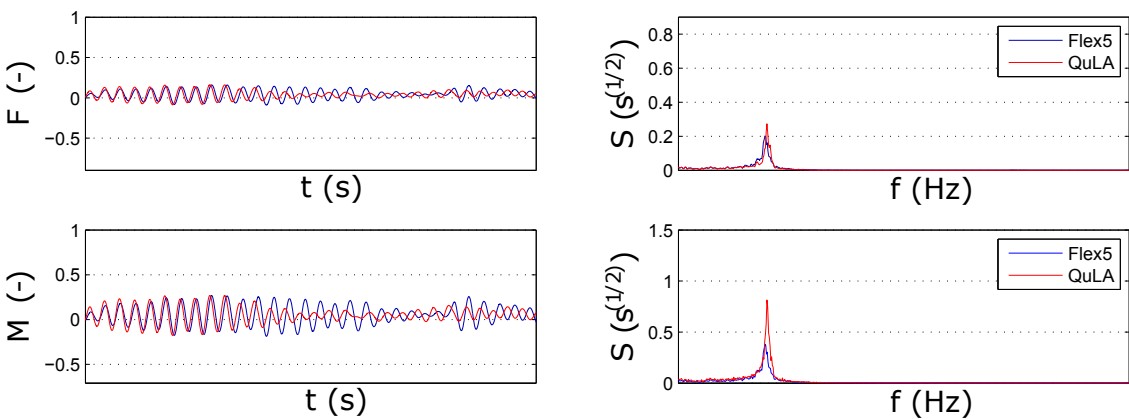

**Figure 19.** Sectional force and moment 26 m above SWL for load case 6.1. The force and moment time series are normalized with the largest force and moment at the sea bed in the Flex5 calculations.

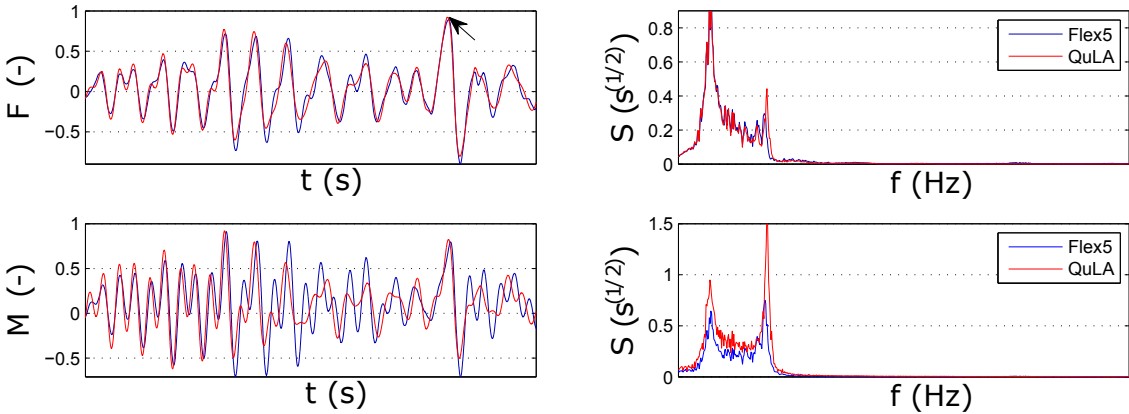

**Figure 20.** Sectional force and moment at sea bed for load case 6.1. The force and moment time series are normalized with the largest force and moment at the sea bed in the Flex5 calculations.

The ratios of the ultimate loads of QuLA to those of Flex5 are seen in figure 21. In the top of the tower the ultimate sectional forces in QuLA are largest with a ratio of 1.04 while just above still water level the two models result in the same ultimate sectional force. Around still water level there is an increase in the difference between the two models and the ratio of the ultimate sectional forces of QuLA to those of Flex5 reduces to 0.7. This is due to Wheeler stretching not applied in QuLA. However, the difference between the models decreases down through the Mono Bucket and at the sea bed the models are very close to each other with a ratio of 0.99. This is expected, since the wave force in load case 6.1 is the largest contributor to the sectional force, and the force at the seabed are the sum of the distributed force, which is calculated in same way in the two models, though not distributed equally.

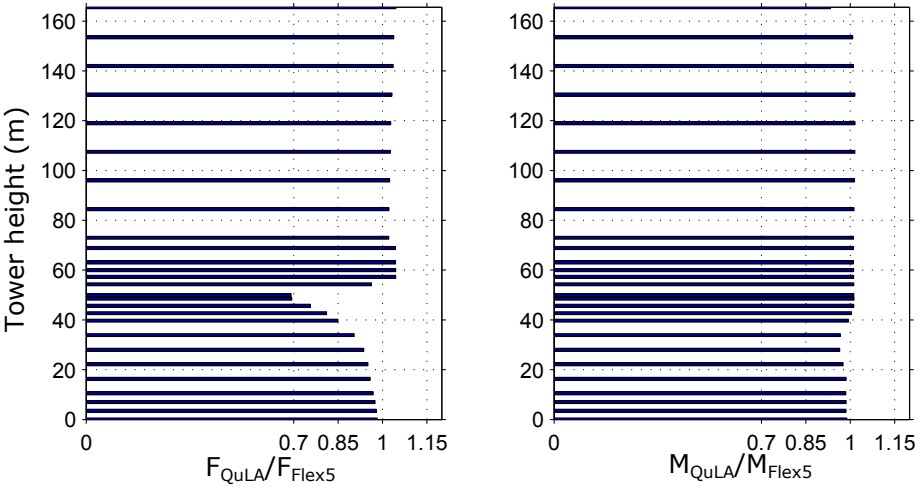

**Figure 21.** The ratio of the ultimate loads of QuLA to those of Flex5 for load case 6.1 in all sections in the tower and Mono Bucket.

With a ratio of approximately 1.02 the difference between the ultimate sectional moments of the two models is more or less constant in all sections in the tower, with those of QuLA being largest. In the Mono Bucket the difference between the two models increases a little just below still water level due to the missing Wheeler stretching in QuLA. However, at the sea bed the ratio is 0.99.

## 5   Conclusions

A model, QuLA, to make fast linear response calculations of the foundation and tower of an offshore wind turbine has been presented. The model solves the equation of motion in the frequency domain and uses precalculated aerodynamic forces and damping as function of the wind speed. Two methods to calculate the aerodynamic damping to be used in QuLA were presented: One based on decay tests calculated in Flex5 and one where the target was to have the same standard deviation of the tower top displacement in Flex5 and QuLA. The damping based on decay tests was found to give the best agreement with Flex5.

To investigate the performance of QuLA the model was compared to Flex5. The shape function and the first natural frequency of the two models are almost identical.

In the fatigue analysis with a decay based damping, the ratio of the equivalent forces of QuLA to those of Flex5 was found to be 0.95 in the tower, while the excitation of the second structural frequency in Flex5 results in larger difference in the Mono Bucket. At the sea bed the equivalent forces of QuLA are smallest with a ratio of 0.7. Considering the equivalent moments, which are often more important, the values of QuLA vary with a ratio between 0.95 and 1.05 to those of Flex5. It would be possible to include a second degree of freedom in QuLA. This will improve the results of QuLA but will essentially also double the complexity of the model.

In the ultimate load analysis, the ratio of both ultimate forces and moments varies between 0.98 and 1.04 in most sections. This difference is due to differences in the dynamic response of the two models and shows that for ULS not only the extreme waves but also the dynamics of the structure is important. At mean water level, though, the missing Wheeler stretching in QuLA, results in much smaller ultimate forces. This difference could be improved by including Wheeler stretching in the model, which though would decrease the computational speed of the model. For the sectional moment in the bottom of the monobucket, however, the agreement was found to be within 2 %.

The proposed model of this paper presents a fast model with good accuracy, especially for the sectional moments. The analysis indicates that in the early stage of the design phase a simple dynamic model can be used in the iterative process to make a preliminary design of the foundation and wind turbine tower. After this, a full aeroelastic model can be used to verify the design and optimize it further. Combined use of a fast and an accurate model might even be applied to enhance this optimization further.

## Acknowledgments

This research was carried out as part of the CEUF-project, (Cost-Effective mass production of Universal Foundations for large offshore wind park), funded by a research project grant from Innovation Fund Denmark. This support is gratefully acknowledged.

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
