# Peer review of "A model for quick load analysis for monopile-type offshore wind turbine substructures"

_Wind Energy Science, 2017_

## Short Comment (SC1) · 10 Mar 2017

It seems the be a very efficient but valid approach for monopiles. The nonlinear wave forcing included with the frequency-domain model shows to give a good ULS estimate. The aerodynamic damping approach also seems to give reasonable results but it will be interesting to see where the differences for the two methods stem from.

Please also note the supplement to this comment:
http://www.wind-energ-sci-discuss.net/wes-2017-11/wes-2017-11-SC1-supplement.pdf

[Figure]

**Supplement:**

[revised manuscript text omitted]

Maero FaeroMaero Iт top xN Faero gMtop faero faero äøm gm wave fwave

Figure 2. Left: Sketch of the beam and the external forces. Right: The external and internal forces which contribute to sectional force, F, and moment, M.

**2.1 The external forces**

10

The external forces are the distributed wave force and the turbulent wind force as seen in figure 2. The pre-calculated rotor
shaft loads are applied as a time varying point force,
$$F_{aero}$$
 and overturning moment,  $M_{aero}$  at the top of the tower. The force
from the wind on the tower is also included and is calculated inside QuLA by the power law from IEC61400-3 IEC61400-3
(2009)

$$f_{aero}(x,t) = \frac{1}{2}\rho_a C_{Da} D\left(\left(\frac{x}{x_n}\right)^{\lambda} W(t)\right)^2,\tag{1}$$

with  $\lambda = 0.14$  in load case 1.2 and load case 1.4 and  $\lambda = 0.11$  in load case 6.1. Here  $\rho_a = 1.225$  kg/m3 is the density of air, 15  $C_{Da} = 0.6$  is the drag coefficient, D(x) is the diameter of the tower and W is the turbulent wind speed at the nacelle.

The wave kinematics and hydrodynamic force are also calculated inside QuLA. To enable fast calculations of the structural response no stretching of the wave kinematics is applied and the wave kinematics are therefore only defined up to still water level, *SWL*.

In situations where fatigue loads are considered, linear wave theory is often sufficient to describe the wave kinematics, 5 Schløer et al. (2016). An irregular wave realization is characterised by the significant wave height  $H_s$  and the peak wave period  $T_p$ . The linear irregular wave kinematics are calculated in the frequency domain and afterwards transformed to the time domain using inverse Fast Fourier transformation. The distributed hydrodynamic load on the structure is calculated by Morison's equation

$$f_{wave}(x,t) = \rho C_m A \dot{u} + \rho A \dot{u} + \frac{1}{2} \rho C_D D u |u|$$
(2)

10 Here ρ = 1025 kg/m3 is the density of water, A(x) is the cross sectional area of the pile and D(x) is the diameter of the pile. The horizontal particle velocity and acceleration are denoted u and u = du/dt. The coefficients, CD and Cm, are the drag and added mass coefficients, with CM = 1 + Cm being the inertia coefficient. The coefficients are functions of the Keulegan-Carpenter number, KC, and Reynolds number, Re, and are calculated following the recommendations in DNV-OS-J101 (2010). For irregular wave realizations KC and Re can, according to Sumer and Fredsøe (2006), be calculated from the standard deviation of the horizontal velocity at still water level and the mean wave period.

The hydrodynamic damping due to the structural motion is considered small and neglected. Therefore, it is not the relative accelerations and the relative velocities, which are considered in the added mass and drag force, first and third term in (2), respectively.

The added mass coefficient,  $C_m$ , is corrected for diffraction effects by the theory of MacCamy-Fuchs, MacCamy and Fuchs 20 (1954), which is valid for linear waves. The correction is important for waves with D/L > 0.2, where L is the wave length. In a water depth of 50m it corresponds to wave frequencies larger than approximately f > 0.19 Hz. To include the diffraction effect, the added mass force is calculated in the frequency domain and afterwards transformed to the time domain.

In order to simultaneously include both the effect of wave irregularity and wave nonlinearity in the structural analysis, IEC61400-3 IEC61400-3 (2009) suggests to embed a large nonlinear stream function wave in the linear irregular wave time

25 series to represent extreme waves. This is done in situations where ultimate loads (ULS) are considered. Following the work of Rainey, Rainey (1989) and Rainey (1995), the Morison's equation is extended by the axial divergence correction term

$$f_{Rainey}(x,t) = \rho A C_m w_x u,\tag{3}$$

which according to Manners and Rainey, Manners and Rainey (1992), corrects for the assumption that the cylinder is slender in the vertical direction. Here the vertical particle velocity is denoted w and index "x" means that the variable is differentiated
30 with respect to x.

Finally a point force should according to Rainey Rainey (1995) be added at the intersection with the water level

$$F_s(t) = -\frac{1}{2}\rho A C_m \eta_z u^2.$$

Here  $\eta_z$  is the slope of the free surface elevation and represents the change of the free surface elevation along the pile-diameter. This force can be seen as a slamming force.

5 The Rainey terms, (3) and (4), are nonlinear contributions to the Morison force and therefore they should only be added to the Morison's equation (2) in situations where a nonlinear single wave event is embedded in the irregular linear wave realization in the ULS-analysis.

**2.2 The structural model**

The structural dynamic deflection of the Mono Bucket and tower, u, is represented by a shape function,  $\varphi$  and a generalized 10 coordinate  $\alpha$  as  $u = \alpha(t)\varphi(x)$ . Shape functions are often introduced when the equation of motion of a system is solved to decrease the number of degrees of freedom in the system and thereby the computational time. Only one shape function is considered in QuLA. While this may not provide an accurate representation of the full deformation, it is here used for the purpose of approximating the associated inertia loads for the sectional forces, see (13)-(14). The shape function and the natural angular frequency,  $\omega_0$  are found by considering a standard eigenvalue problem,

15
$$\mathbf{M}\ddot{\alpha}\underline{\varphi} + \mathbf{K}\alpha\underline{\varphi} = 0$$
, where  $\alpha = \exp(i\omega_0 t) \Leftrightarrow$  (5a)
 $-\mathbf{M}\omega_0^2\varphi + \mathbf{K}\varphi = 0 \Rightarrow \omega_0^2\varphi = \mathbf{M}^{-1}\mathbf{K}\varphi.$  (5b)

The stiffness and mass matrix is calculated by the finite element method. Stiffness elements representing the stiffness from the soil-structure interaction,  $K_s$  in figure 2, is calculated in the geotechnical software tool Plaxis, Brinkgreve et al. (2016) and is added to the stiffness matrix in the bottom of the pile. The top mass and mass moment of inertia around the nacelle (y-axis),  $I_T$ , are added to the mass matrix in the top of the pile. To get the correct first natural frequency it is important to define  $M_{top}$

20

and  $I_T$  in same height as the center of mass in the nacelle,  $x_N$ .

The structural dynamics are calculated by the equation of motion

$$\ddot{\alpha}GM + \alpha GK + \dot{\alpha}GD = GF. \tag{6}$$

In order for the model to be fast the equation of motion is solved in frequency domain, since the solution  $\alpha$  can then be solved at once for all time steps. In frequency domain the generalized coordinate can be expressed as

$$\bigotimes_{j=1}^{N_f} \hat{\alpha}_j \exp(i\omega_j t) + c.c., \tag{7}$$

(4)

where  $\omega_j$  is the smallest angular frequency in the time series and c.c. is the complex conjugate. The equation of motion

$$-\omega^2 G M \hat{\alpha} + i\omega G D \hat{\alpha} + G K \hat{\alpha} = G F \Leftrightarrow \hat{\alpha} = \frac{\hat{G} F}{-\omega^2 G M + i\omega G D + G K}$$
(8)

then solves the linear response in frequency domain and can readily be transposed to the time domain by inverse FFT.

The generalized mass, GM, and stiffness, GK, can be obtained from (5a) by left-multiplication of  $\varphi^{T}$  or are given as

5
$$GM = \int_{x=0}^{x_{TT}} m\varphi(x)^2 dx + M_{top}\varphi(x_N)^2 + I_T\varphi_x(x_N)^2,$$
 (9)

$$GK = \int_{x=0}^{x_{TT}} EI\varphi_{xx}(x)^2 \, dx.$$
 (10)

Here m(x) is the distributed mass of the tower and Mono Bucket foundation,  $\varphi_x$  is the angular deflection of the shape function and  $\varphi_{xx}$  is the curvature of the shape function. The stiffness factor is given by the modulus of elasticity, E, and the moment of inertia I. Further, the damping, GD, and force, GF, are given as

$$\quad GD = \zeta \frac{2GK}{\omega_0} + D_{aero}, \tag{11}$$

$$GF = \int_{x=0}^{x_{SWL}} \varphi f_{wave} dx + F_s + F_{aero} \varphi(x_{TT}) + M_{aero} \varphi_x(x_{TT}) + \int_{x_{SWL}}^{x_{TT}} \varphi f_{aero} dx, \tag{12}$$

The damping,  $\zeta$ , is the damping ratio representing structural damping, soil damping and hydrodynamic radiation damping and  $D_{aero}$  the aerodynamic damping.

After the equation of motion is solved, the sectional forces and moments can be calculated. The external and internal forces, which contribute to the sectional forces and moments are shown in figure 2 and the forces and moments are calculated as

$$F(x^{*},t) = -\ddot{\alpha} \int_{x^{*}}^{x_{TT}} m\varphi(x)dx - \ddot{\alpha}M_{top}\varphi(x_{N}) + \int_{x^{*}}^{x_{SWL}} f_{wave}dx + F_{s} + F_{aero} + \int_{x^{*}}^{x_{TT}} f_{aero}dx + \alpha g M_{top}\varphi_{x}(x_{N}) + \alpha g \int_{x^{*}}^{x_{TT}} m\varphi_{x}(x)dx$$

$$M(x^{*},t) = -\ddot{\alpha} \int_{x^{*}}^{x_{TT}} m\varphi(x)[x - x^{*}]dx - \ddot{\alpha}M_{top}\varphi(x_{N})[x_{n} - x^{*}] - \ddot{\alpha}I_{T}\varphi_{x}(x_{N}) + \int_{x^{*}}^{x_{SWL}} f_{wave}[x - x^{*}]dx + F_{s}[x_{SWL} - x^{*}] + M_{aero} + F_{aero}[x_{TT} - x^{*}] + \int_{x^{*}}^{x_{TT}} f_{aero}[x - x^{*}]dx + \alpha M_{top}g[\varphi(x_{N}) - \varphi(x^{*})] + \alpha g \int_{x^{*}}^{x_{TT}} m[\varphi(x_{TT}) - \varphi(x)]dx,$$
(13)
(13)

5

where g is the gravity. The first two terms in both equations are the contribution from the dynamics of the structure. When the equation of motion is solved the Mono Bucket and tower are treated as an Euler beam, where the deflections are assumed small and only lateral loads are considered. Second-order contributions from the bending of the beam are therefore neglected in the solution in order for the model to be fast. However, in the sectional forces and moment the contribution from gravity due to the bending of the beam is included as stated in the last two terms in both equations. While this approach thus represent a difference in the forces applied for dynamics and sectional loads, it was found to improve the sectional loads.

10

**2.3 Shape function and eigenfrequency**

The complete shape function of both the tower and bucket foundation in Flex5 is compared to the shape function of QuLA in figure 3. The shape functions are close to being identical. The deviation between the first natural frequency of the two models 15 is 1%. The difference is caused by differences in the models: In Flex5 the gravity's contribution to the bending of the pile is included in the equation of motion, which gives a larger moment of inertia and therefore a smaller frequency. In QuLA the contribution of the gravity is only included in the sectional forces calculated after the equation of motion is solved.

**2.4 The aerodynamic damping**

As the structural dynamics is included in QuLA, it is also necessary to include the aerodynamic damping. If the structural motion is in same direction as the wind velocity, the relative velocity which the aerodynamic forces are a function of, decreases and thereby also the forces. Since the aerodynamic forces are included as point forces in QuLA, it is necessary to simplify the aerodynamic damping and add the damping to the equation of motion as a viscous linear damping force, where the damping coefficient is a function of the mean wind speed.

Two different methods to calculate the damping are presented below and compared for load case 1.2 in section 4.1.

---

## Referee Comment (RC1) · Anonymous Referee #1 · 23 Mar 2017

- It is important to highlight the issues and aspects of the "quick" load calculation method that have been addressed here and have not been considered before.

- Does the overturning moment include both the fore-aft and side-side bending moments.

- please elaborate on how the aerodynamic damping has been precalculated.

- please consider the use of a list of symbols, since symbols used in the figures are not self explanatory.

- some information about the wind turbine would be useful since some parameters are mentioned many time such as, rated wind speed, rotational speed at rated etc.

- the damping behavior may change with the turbulence intensity, especially when the wind turbine is operating above rated wind speed, would the computed damping from the different methods change with turbulence intensity.

- the most critical state is usually around rated wind speed. In this case the wind speed fluctuate between below rated (where the pitch is not active) to above rated where the pitch is activated. How would this constant switching behavior affect the calculation of the damping ratio?

- please be more specific on the loads that have been calculated with this method, bending moment in both directions?

- for the design of monopile/bucket, pile rotation could be important

[revised manuscript text omitted]

λ, wind shear exponent,
usually it is designated as
α, since λ is widely used
as the tip speed ratio.

[Figure]

[Figure]

The wave kinematics and hydrodynamic force are also calculated inside QuLA. To enable fast calculations of the structural response no stretching of the wave kinematics is applied and the wave kinematics are therefore only defined up to still water level, $SWL$.

In situations where fatigue loads are considered, linear wave theory is often sufficient to describe the wave kinematics, Schløer et al. (2016). An irregular wave realization is characterised by the significant wave height $H_s$ and the peak wave period $T_p$. The linear irregular wave kinematics are calculated in the frequency domain and afterwards transformed to the time domain using inverse Fast Fourier transformation. The distributed hydrodynamic load on the structure is calculated by Morison's equation

$$f_{wave}(x,t) = \rho C_m A \dot{u} + \rho A \dot{u} + \frac{1}{2}\rho C_D D u\, u \tag{2}$$

Here $\rho - 1025$ kg/m$^3$ is the density of water, $A(x)$ is the cross sectional area of the pile and $D(x)$ is the diameter of the pile. The horizontal particle velocity and acceleration are denoted $u$ and $\dot{u} = \frac{du}{dt}$. The coefficients, $C_D$ and $C_m$, are the drag and added mass coefficients, with $C_M = 1 + C_m$ being the inertia coefficient. The coefficients are functions of the Keulegan-Carpenter number, $KC$, and Reynolds number, $Re$, and are calculated following the recommendations in DNV-OS-J101 (2010). For irregular wave realizations $KC$ and $Re$ can, according to Sumer and Fredsøe (2006), be calculated from the standard deviation of the horizontal velocity at still water level and the mean wave period.

The hydrodynamic damping due to the structural motion is considered small and neglected. Therefore, it is not the relative accelerations and the relative velocities, which are considered in the added mass and drag force, first and third term in (2), respectively.

The added mass coefficient, $C_m$, is corrected for diffraction effects by the theory of MacCamy-Fuchs, MacCamy and Fuchs (1954), which is valid for linear waves. The correction is important for waves with $D/L > 0.2$, where $L$ is the wave length. In a water depth of 50m it corresponds to wave frequencies larger than approximately $f > 0.19$ Hz. To include the diffraction effect, the added mass force is calculated in the frequency domain and afterwards transformed to the time domain.

In order to simultaneously include both the effect of wave irregularity and wave nonlinearity in the structural analysis, IEC61400-3 IEC61400-3 (2009) suggests to embed a large nonlinear stream function wave in the linear irregular wave time series to represent extreme waves. This is done in situations where ultimate loads (ULS) are considered. Following the work of Rainey, Rainey (1989) and Rainey (1995), the Morison's equation is extended by the axial divergence correction term

$$f_{Rainey}(x,t) = \rho A C_m w_x u. \tag{3}$$

which according to Manners and Rainey, Manners and Rainey (1992), corrects for the assumption that the cylinder is slender in the vertical direction. Here the vertical particle velocity is denoted $w$ and index "$x$" means that the variable is differentiated with respect to $x$.

[Figure]

[Figure]

Finally a point force should according to Rainey Rainey (1995) be added at the intersection with the water level

$$F_s(t) = -\frac{1}{2}\rho A C_m \eta_z u^2. \tag{4}$$

*in the Z direction where the slope is being calculated, the same as the X direction used before?*

Here $\eta_z$ is the slope of the free surface elevation and represents the change of the free surface elevation along the pile-diameter. This force can be seen as a slamming force. *why can $C_m$ be used instead of a slamming coefficient?*

5 The Rainey terms, (3) and (4), are nonlinear contributions to the Morison force and therefore they should only be added to the Morison's equation (2) in situations where a nonlinear single wave event is embedded in the irregular linear wave realization in the ULS-analysis.

**2.2 The structural model**

The structural dynamic deflection of the Mono Bucket and tower, u, is represented by a shape function, $\varphi$ and a generalized
10 coordinate $\alpha$ as $u = \alpha(t)\varphi(x)$. Shape functions are often introduced when the equation of motion of a system is solved to decrease the number of degrees of freedom in the system and thereby the computational time. Only one shape function is considered in QuLA. While this may not provide an accurate representation of the full deformation, it is here used for the purpose of approximating the associated inertia loads for the sectional forces, see (13)-(14). The shape function and the natural angular frequency, $\omega_0$ are found by considering a standard eigenvalue problem,

$$\mathbf{M}\ddot{\alpha}\underline{\varphi} + \mathbf{K}\alpha\underline{\varphi} = 0, \quad \text{where} \quad \alpha = \exp(i\omega_0 t) \Leftrightarrow \tag{5a}$$

$$-\mathbf{M}\omega_0^2\underline{\varphi} + \mathbf{K}\underline{\varphi} = 0 \Rightarrow \omega_0^2\underline{\varphi} = \mathbf{M}^{-1}\mathbf{K}\underline{\varphi}. \tag{5b}$$

The stiffness and mass matrix is calculated by the finite element method. Stiffness elements representing the stiffness from the soil-structure interaction, $K_s$, in figure 2, is calculated in the geotechnical software tool Plaxis, Brinkgreve et al. (2016) and is added to the stiffness matrix in the bottom of the pile. The top mass and mass moment of inertia around the nacelle ($y$-axis),
20 $I_T$, are added to the mass matrix in the top of the pile. To get the correct first natural frequency it is important to define $M_{top}$ and $I_T$ in same height as the center of mass in the nacelle, $x_N$.

The structural dynamics are calculated by the equation of motion

$$\ddot{\alpha}GM + \alpha GK + \dot{\alpha}GD = GF. \tag{6}$$

*G denotes generalized? a is G a matrix as well?*

In order for the model to be fast the equation of motion is solved in frequency domain, since the solution $\alpha$ can then be
25 solved at once for all time steps. In frequency domain the generalized coordinate can be expressed as

$$\alpha = \sum_{j=1}^{N_f} \hat{\alpha}_j \exp(i\omega_j t) + c.c., \tag{7}$$

*Why not use standard symbol like — to denote conjugate complex?*

[Figure]

where $\omega_j$ is the smallest angular frequency in the time series and $c.c.$ is the complex conjugate. The equation of motion

$$-\omega^2 GM\hat{\alpha} + i\omega GD\hat{\alpha} + GK\hat{\alpha} = GF \Leftrightarrow \hat{\alpha} = \frac{\hat{G}F}{-\omega^2 GM + i\omega GD + GK} \tag{8}$$

then solves the linear response in frequency domain and can readily be transposed to the time domain by inverse FFT.

The generalized mass, $GM$, and stiffness, $GK$, can be obtained from (5a) by left-multiplication of $\varphi^{\mathrm{T}}$ or are given as

$$\quad GM = \int_{x=0}^{x_{TT}} m\varphi(x)^2 \, dx + M_{top}\varphi(x_N)^2 + I_T\varphi_x(x_N)^2, \tag{9}$$

$$GK = \int_{x=0}^{x_{TT}} EI\varphi_{xx}(x)^2 \, dx. \tag{10}$$

Here $m(x)$ is the distributed mass of the tower and Mono Bucket foundation, $\varphi_x$ is the angular deflection of the shape function and $\varphi_{xx}$ is the curvature of the shape function. The stiffness factor is given by the modulus of elasticity, $E$, and the moment of inertia $I$. Further, the damping, $GD$, and force, $GF$, are given as

$$\quad GD = \zeta \frac{2GK}{\omega_0} + D_{aero}, \tag{11}$$

$$GF = \int_{x=0}^{x_{SWL}} \varphi f_{wave} dx + F_s + F_{aero}\varphi(x_{TT}) + M_{aero}\varphi_x(x_{TT}) + \int_{x_{SWL}}^{x_{TT}} \varphi f_{aero} dx, \tag{12}$$

The damping, $\zeta$, is the damping ratio representing structural damping, soil damping and hydrodynamic radiation damping and $D_{aero}$ the aerodynamic damping.

After the equation of motion is solved, the sectional forces and moments can be calculated. The external and internal forces,
15 which contribute to the sectional forces and moments are shown in figure 2 and the forces and moments are calculated as

$$F(x^*,t) = -\ddot{\alpha} \int_{x^*}^{x_{TT}} m\varphi(x)dx - \ddot{\alpha}M_{top}\varphi(x_N) + \int_{x^*}^{x_{SWL}} f_{wave}dx + F_s + F_{aero}$$

$$+ \int_{x^*}^{x_{TT}} f_{aero}dx + \alpha g M_{top}\varphi_x(x_N) + \alpha g \int_{x^*}^{x_{TT}} m\varphi_x(x)dx \tag{13}$$

$$M(x^*,t) = -\ddot{\alpha} \int_{x^*}^{x_{TT}} m\varphi(x)[x-x^*]dx - \ddot{\alpha}M_{top}\varphi(x_N)[x_n-x^*] - \ddot{\alpha}I_T\varphi_x(x_N)$$

$$+ \int_{x^*}^{x_{SWL}} f_{wave}[x-x^*]dx + F_s[x_{SWL}-x^*] + M_{aero} + F_{aero}[x_{TT}-x^*]$$

$$+ \int_{x^*}^{x_{TT}} f_{aero}[x-x^*]dx + \alpha M_{top}g[\varphi(x_N)-\varphi(x^*)] + \alpha g \int_{x^*}^{x_{TT}} m[\varphi(x_{TT})-\varphi(x)]dx, \tag{14}$$

where $g$ is the gravity. The first two terms in both equations are the contribution from the dynamics of the structure. When the equation of motion is solved the Mono Bucket and tower are treated as an Euler beam, where the deflections are assumed small and only lateral loads are considered. Second-order contributions from the bending of the beam are therefore neglected in the solution in order for the model to be fast. However, in the sectional forces and moment the contribution from gravity due to the bending of the beam is included as stated in the last two terms in both equations. While this approach thus represent a difference in the forces applied for dynamics and sectional loads, it was found to improve the sectional loads.

**2.3 Shape function and eigenfrequency**

The complete shape function of both the tower and bucket foundation in Flex5 is compared to the shape function of QuLA in figure 3. The shape functions are close to being identical. The deviation between the first natural frequency of the two models is 1%. The difference is caused by differences in the models: In Flex5 the gravity's contribution to the bending of the pile is included in the equation of motion, which gives a larger moment of inertia and therefore a smaller frequency. In QuLA the contribution of the gravity is only included in the sectional forces calculated after the equation of motion is solved.

**2.4 The aerodynamic damping**

As the structural dynamics is included in QuLA, it is also necessary to include the aerodynamic damping. If the structural motion is in same direction as the wind velocity, the relative velocity which the aerodynamic forces are a function of, decreases and thereby also the forces. Since the aerodynamic forces are included as point forces in QuLA, it is necessary to simplify the aerodynamic damping and add the damping to the equation of motion as a viscous linear damping force, where the damping coefficient is a function of the mean wind speed.

Two different methods to calculate the damping are presented below and compared for load case 1.2 in section 4.1.

[Figure]

[Figure]

[Figure]

**Figure 3.** The shape function.

**2.4.1 Standard deviation of pile displacement**

In this approach the target is to have the same standard deviation of the pile displacement in the top of the tower. Therefore the tower top displacement has to be calculated in advance in Flex5 or another aeroelastic model for all considered cases. In QuLA, when the equation of motion is solved, a loop is included, where the aerodynamic damping is increased until the standard deviation is the same for Flex5 and QuLA. The standard deviation is calculated as

$$\sigma = \sqrt{\frac{1}{2} \sum \hat{u}(x_{NN})^2 \Delta f}.$$  (15)

where $\hat{u}(x_{NN})$ is the tower top displacement in frequency domain.

In figure 4-5 the tower top displacements calculated in Flex5 and QuLA for $W = 4.16$m/s and $W = 14.55$m/s are shown for load case 1.2. For the small wind speed the two models compare very well, however as the wind increases differences between the two models are seen. This is due to differences in how the model is solved. In Flex5 the aerodynamic damping is a function of time, while in QuLA it is represented by a constant value for each wind speed. Further, in QuLA only one degree of freedom is used and the gravity's contribution to the deflection is not included in QuLA as mentioned in section 2.3.

**2.4.2 Decay tests**

The amount of damping, which should be included, is calculated in Flex5 by decay tests. To calculated the damping both turbulent and steady wind speeds are considered. For both cases two simulations are run. One where a starting velocity of the

② why pile displacement, why not velocity, as the aerodynamic damping is ⁸proportional to the Tower Top velocity

③ more visible?

④ How does a decay test looks like with a rotating rotor?

[Figure]

[Figure]

[Figure]

**Figure 4.** Tower top displacement for $W = 4.16$ m/s.

**Figure 5.** Tower top displacement for $W = 14.55$ m/s.

*afterwards,*

wind turbine tower and foundation is applied and one simulation without a starting velocity. Afterwords the two simulations are subtracted before the damping is calculated.

All degrees of freedom are active, however the rotor speed is kept constant and the pitch angle and rotational speed of the blades are given initial values in accordance with the wind speeds considered. According to Salzmann and Van der Tempel

5  (2005) this method works well for constant speed wind turbines and compares well with other simple methods as the Garrad method Freris and Freris (1990), Kühn's closed-form model Kühn (2001) or van der Tempels method, Van Der Tempel (2006). However, for a pitch regulated wind turbine with varying rotor speed, which is the case for the DTU 10MW wind turbine, such simple methods can not be applied to find the accurate damping above rated wind speed, where the pitch regulation begins. However, the damping in Qula can only be represented by a single value as function of the mean wind speed. Therefore, the

10  damping above rated wind speed is still found by keeping the pitch and rotor speed constant, since it is a very simple method which can be reused several times as long as the wind turbine is the same.

The logarithmic decrement damping is calculated as

$$\delta = \frac{1}{j} \log\left(\frac{\hat{a}_1}{\hat{a}_j}\right), \quad \text{where} \quad j = 2, 3.....$$  (16)

where $\hat{a}_1$ is the first peak considered in the time series and $\hat{a}_j$ is the $j$'th amplitude following $\hat{a}_1$. The relation between the

15  logarithmic decrement, damping ratio, $\xi$ and the damping which is used in the dynamic analysis $d$ is

$$\delta = \frac{2\pi\xi}{\sqrt{1 - \xi^2}};$$  (17)

*are there the same?*

$$D_{aero} = \zeta 2\sqrt{GM\,GK};$$  (18)

*please describe the symbol which is not used before*

where $GM$ and $GK$ are the generalised stiffness and mass, cf. section 2.2.

In figure 6-7 the decay tests for a steady and turbulent wind speed of 14 m/s are shown. In the top figures the displacements

20  in the top of the tower are shown both for the case where the tower has an initial velocity of $U_{init} \sim 1.1$m/s and the one without

① what does subtracted mean here?

②③ what is the difference between rotor speed and the rotational speed of the blades?

④ please justify why this assumption is still acceptable

[Figure]

an initial velocity and in the bottom the subtracted displacements are shown. The logarithmic decrement is the average of the four peaks following the largest peak, and is calculated for both the negative and positive peaks.

[Figure]

**Figure 6.** Decay test for a steady wind of 14 m/s.

[Figure]

**Figure 7.** Decay test for a turbulent wind of 14 m/s.

In figure 8 the damping ratios as function of both steady and turbulent wind speed are shown for three initial tower velocities. In the figure the average of the six curves is also seen. It is seen that the damping ratio is very similar across the initial tower

5  velocities.

[Figure]

**Figure 8.** The damping ratio as function of wind speed for different decay-tests.

**2.4.3  Comparison of the damping**

In figure 9 the damping ratio as function of the wind speed from cut-in to cut-out wind speed is shown for the two different methods to calculate the damping. It is seen that the damping based on decay tests is larger than when it is based on the standard

deviation of the tower displacement except for $W \sim 17$ m/s, where the damping based on the tower top displacement is largest. However, the trend is similar for both damping curves.

The damping is constant for small wind speeds with a value between 7-8 % for the decay tests and 2 % for the tower top displacements but starts to increase before rated wind speed.

5     For the decay tests the largest damping ratio of 10.5 % is reached for a wind speed of 12 m/s. Above rated wind speed, the damping decreases and is approximately 9% for a wind speed of 25 m/s.

For wind speeds between 10 and 17 m/s, the damping ratio based on the tower top displacements increases from 2-10.3% , where after it decreases and is 7% for a wind speed of 25 m/s.

[Figure]

**Figure 9.** The damping ratio as function of wind speed for the two methods to calculate the damping.

Both methods require some preliminary work to calculate the viscous damping, to be used in QuLA. Either decay tests have

10     to be made or the displacement in the top of the tower has to be calculated in an aeroelastic tool. However, as the foundation is very stiff, it is not believed that the foundation contributes significant to the damping. Therefore, the preliminary work can be made for a land based wind turbine, and the aerodynamic damping reused several times as long as the wind turbine and tower is not changed.

How the different damping curves influence the performance of QuLA is investigated in section 4.1.

① this is probably true as long as the wind turbine faces free stream wind speed, what about wake situation?

15    **3   Metocean data and structure**

The load cases in the present analysis are based on the metocean data from the artificial site "'K13 Deepwater Site'" from the Upwind-project Fischer et al. (2010). The water depth is $h = 50$m. Three load cases are studied, load case 1.2 which consider the fatigue limit state (FLS) and load case 1.3 and 6.1 which consider the ultimate limit state (ULS). The time series of each wind and sea state is 1 hour long which corresponds to six seeds of 600 s. In load case 1.2 the wind turbine operates, and

② the contribution of the soil to the total damping depends strongly on the soil properties. Stiff soils such as the North sea yield very little damping. Cohesive soils with weak stiffness may contribute to the damping that cannot be neglected.

[Figure]

[Figure]

the wind speed ranges from 4m/s to 25 m/s with an interval of 2m/s using a normal turbulence model. The already lumped sea states presented in Fischer et al. (2010) is used together with the wind speeds. Since fatigue loads are considered the wind speeds probability of occurrence is taken into account.

The wind speeds and the corresponding probability of occurrence, $P_r$, turbulence intensity, $I$, and sea states are stated in
5  table 1.

In load case 1.3 the wind turbine also operates and the wind speed are the same as load case 1.2, but the turbulence intensity is now based on an extreme turbulence model. The significant wave height is the expected wave height conditioned on the wind speed

$$H_s = E[H_S \, V_{hub}] = \sum_i H_{s,i} P_{rel}, \tag{19}$$

10  where $P_{rel}$ is the relative probability of occurrence of each significant wave height conditioned on the considered wind speed. The range of peak wave periods appropriate to each $H_s$ should be taken into account and the one resulting in the largest load should be used in the ULS-analysis. Further, if the peak wave period corresponding to the first natural frequency, $f_1 = 1/Tp$ is inside the considered range this wave period should also be considered. The same applies to higher hormonics of the wave peak period, i.e. multiple of the peak wave frequency, $2f_p$ and $3f_p$, as this will cause a a larger excitation of the structure. In the
15  present analysis , the largest and smallest wave peak period which occur, are considered. The wind speed, turbulent intensity and corresponding $H_s$ and $T_p$ values are stated in table 2. Also the periods in between the smallest and largest $T_p$-value, which frequency or its multiples are equal to the first natural frequency are considered. However, do to confidential design, these frequencies are not written in the table, but a + indicates for which wind speeds they occur.

In load case 6.1 the wind turbine is parked and the wind speed is 44.03 m/s. The corresponding sea state has a significant
20  wave height of $H_s = 9.40$m and a peak period of $T_p = 10.87$s.

In ULS situations a irregular linear wave time series is first created. For every 600 s the largest wave in the interval is replaced with a nonlinear regular stream function wave with a wave height of $H = 1.86H_s$, IEC61400-3 (2009). The corresponding wave period should according to IEC61400-3 (2009), be chosen as the period in the interval

$$11.1\sqrt{H_s/g} < T < 14.3\sqrt{H_s/g}, \tag{20}$$

25  which results in the largest load. For the present structure that is $T = 11.1\sqrt{H_s/g} = 10.87$s in load case 6.1. In load case 1.3 the same ratio, $T = 11.1\sqrt{H_s/g}$ is used.

The wind turbine is the 10 MW DTU reference wind turbine, Bak et al. (2013). The first natural frequency of the structure is in between the 1P and 3P frequency interval of the wind turbine (1P=0.115–0.159 Hz). The Mono Bucket foundation is designed to withstand the extreme static forces stated in the report of the DTU 10 MW wind turbine Bak et al. (2013). In both

[Figure]

| W | $P_r$ | I | $H_S$ | $T_p$ |
|---|---|---|---|---|
| (m/s) | (-) | (-) | (m) | (s) |
| 4.16 | 0.11 | 0.29 | 1.10 | 5.88 |
| 6.23 | 0.14 | 0.23 | 1.18 | 5.76 |
| 8.31 | 0.16 | 0.20 | 1.31 | 5.67 |
| 10.39 | 0.15 | 0.18 | 1.48 | 5.74 |
| 12.47 | 0.13 | 0.17 | 1.70 | 5.88 |
| 14.55 | 0.11 | 0.16 | 1.91 | 6.07 |
| 16.62 | 0.08 | 0.15 | 2.19 | 6.37 |
| 18.70 | 0.05 | 0.15 | 2.47 | 6.71 |
| 20.78 | 0.03 | 0.14 | 2.76 | 6.99 |
| 22.56 | 0.02 | 0.14 | 3.09 | 7.40 |
| 24.94 | 0.01 | 0.14 | 3.42 | 7.80 |

**Table 1.** The wind speeds and the corresponding probability of occurrence, turbulence intensity and sea states and for load case 1.2.

| W | I | $H_S$ | $T_{p,min}$ | $T_{p,max}$ | $T_p$ for $f_1 = 1/Tp$ | $T_p$ for $f_1 = 2/Tp$ |
|---|---|---|---|---|---|---|
| (m/s) | (-) | (-) | (m) | (s) | (s) | (s) |
| 4.16 | 0.82 | 5.88 | 4 | 11 | + | + |
| 6.23 | 0.90 | 5.76 | 4 | 11.5 | + | + |
| 8.31 | 1.05 | 5.67 | 4 | 11.5 | + | + |
| 10.39 | 1.23 | 5.74 | 4 | 11. | + | + |
| 12.47 | 1.46 | 5.88 | 5 | 9 | - | - |
| 14.55 | 1.72 | 6.07 | 5 | 8 | - | - |
| 16.62 | 2.07 | 6.37 | 5 | 9 | - | - |
| 18.70 | 2.38 | 6.71 | 5 | 10 | - | + |
| 20.78 | 2.80 | 6.90 | 5 | 8 | - | - |
| 22.56 | 3.13 | 7.40 | 7 | 9 | - | - |
| 24.94 | 3.58 | 7.80 | 7 | 10 | - | + |

**Table 2.** The wind speeds and the corresponding probability of occurrence, turbulence intensity and sea states and for load case 1.3

Flex5 and QuLA a logarithmic damping of $\delta = 2\pi\zeta = 6\%$ is included as viscous damping 
[revised manuscript text omitted]

*① what is the definition of the ultimate loads?*
*which exceedance probability or return period?*

WIND
ENERGY
SCIENCE
DISCUSSIONS

[Figure]

**Figure 16.** Probability of exceedance of the positive peaks in the time series of the sectional forces and moments for load case 1.3. Blue: Flex5. Red: QuLA

largest ultimate forces due to the excitation of the second natural frequency. The difference between the models increases from the top to the bottom of the Mono Bucket and at the sea bed the ratio is 0.85. The effect of the second natural frequency is also visible in ultimate moments, but the effect is not as strong, since the forces in the tower still contribute more to moment at the sea bed, where the ratio between the two models is 0.95.

[Figure]

**Figure 17.** The ratio of the ultimate loads of QuLA to those of Flex5 for load case 1.3 in all sections in the tower and Mono Bucket.

5  ### 4.2.2  Load case 6.1

Load case 6.1 considers a storm condition. The wind turbine is therefore parked, and the aerodynamic force and damping are therefore small. The contribution from the wave force is therefore expected to be significant.

[Figure]

In figure 18 the probability of exceedance, $P$, of the positive peaks in the 1 hour time series of the sectional forces and moments in five sections of the Mono Bucket and tower are seen.

[Figure]

**Figure 18.** Probability of exceedance of the positive peaks in the time series of the sectional forces and moments for load case 6.1. Blue: Flex5. Red: QuLA

QuLA has the largest force peaks for high probability of exceedance while Flex5 has the largest force peaks for low probability of exceedance, however in the tower the probability of the force peaks are quite equal. At still water level there is a large

5 difference between the curves of the two models, in particular for $P < 0.03$. The peaks of Flex5 is largest, which is caused by the Wheeler stretching in Flex5, which stretches the wave kinematics up to the free surface elevation instead of only being defined to still water level as in QuLA. In the Mono Bucket 26 m above the sea bed, the difference between the two models is still significant but smaller. At the sea bed the force-curves of the two models are again quite equal.

The probability curves of the moments of the two models are more equal in all five sections. Particularly the largest moments,

10 which are important in ULS, compare well. To compare the dynamics of the two models a sequence of the time series and response amplitude spectra of the 1 hour time series of the sectional forces and moments at the intersection between the tower and Mono Bucket and at the sea bed are considered, figures 19-20. In the tower, the energy of the force and moment is located around the first natural frequency, however QuLA contains more energy at this frequency. In the time series, the response *damped* dampens faster in Flex5. 
[revised manuscript text omitted]

---

## Referee Comment (RC2) · M Muskulus (Referee) · 24 Apr 2017

The manuscript proposes a simplified wind turbine load analysis method for windturbines on monopiles that is based on a single degree of freedom model of the substructure. Hydrodynamic loads are integrated with the shape function, resulting in a generalized load. Wheeler stretching has not been applied. Aerodynamic loads are pre-computed on a rigid turbine and are applied as point force and momentum time series. The aerodynamic damping is determined either by matching the standard deviation of tower top displacements, or by a decay test performed. The latter method determines the damping ratio from the observed decay in the difference of the response with an initial tower top velocity and a tower initially at rest. It is considered by the authors to be more accurate. The equation of motion is solved in the frequency domain, and results are presented for the DTU 10MW reference wind turbine on a

novel Mono Bucket concept. Compared with state of the art FLEX5 simulations of the same turbine, fatigue damage is underestimated in the Mono Bucket, with a damage equivalent load of around 70 percent of the FLEX5 result. The ultimate loads under operating conditions are likewise underestimated in the Mono Bucket by 15 percent. Under storm conditions with a parked turbine, the biggest differences occur in the ultimate loads close to the waterline, which are underestimated by around 30 percent. This is presumably due to the missing wave stretching.

The paper is mostly well written, although the authors have been a bit careless and many typos exist that should be corrected. The topic is interesting and suitable for Wind Energy Science journal. However, a number of issues should be addressed before publication.

1. New content: As this paper is part of a Special Issue of papers previously published with IOP (from The Science of Making Torque From Wind conference), I would have expected a footnote explaining this fact. As the authors are probably aware of, publication in Wind Energy Science journal is contingent on 40 percent new content compared to the previously published work. Can the authors explain a bit how they have updated the conference paper and what new results/content is included in the manuscript here?

2. Use of frequency domain: It is unclear if the phase information is retained or not. Are the coefficients $\hat{\alpha}_j$ in Eq. 7 complex? The text ("can readily be transposed to the time domain by inverse FFT") suggests this is the case. If so, the solution is completely equivalent to a time domain integration. What is the reason for the use of the frequency domain then? The speedup due to the possibility of using FFT? Please comment and discuss in the text.

3. Aerodynamic damping: p7, l21f: "... it is necessary to simplify the aerodynamic and add the damping [...] as a viscous linear damping force ..." - This seems a bit too suggestive. Why is it "necessary" to model the damping with a linear viscous damper? (In fact, the aerodynamic damping force is definitely not linear)

4. Calculation of standard deviation: Eq. 15 seems to have some issues. First, why the factor of 1/2? What is the summation over? How is the displacement \hat{u} determined from the previous \hat{\alpha} - is it the same? And should it not be an absolute square of the (complex?) displacements?

5. Determination of damping ratio: p9, l9f: "The damping [...] is found by keeping the pitch and rotor speed constant, since it is a very simple method which can be reused several times" - Unclear what the latter part of this sentence refers to. Please explain.

6. Determination of damping ratio: p10, l1f: "The logarithmic decrement is the average of the four peaks ..." - Imprecise formulation. Did you mean: "The logarithmic decrement has been estimated for the four peaks ... and then averaged"?

7. Discussion of damping: In general, the term "damping" is used somewhat ambiguously. Without further explanation, I would assume it stands for a damping force, but the authors seem to use it mostly for the "damping ratio". Please consider a more precise use. Also, p13, l1: It seems unusual to report the logarithmic decrement in percent - percent of what? This is normally used for damping ratios only (percent of critical damping), and therefore misleading here.

8. ULS wave loads: p12, l25: How were the values giving the largest wave loads in the interval from Eq. 20 determined? Values are given in line 25, but how were they found?

9. Focus on speed: The main motivation for the method seems to be that it results in much faster load simulations. However, alternatives exist that should at least be discussed, e.g. the convolution-based approach in the time domain (Schafhirt et al.: "Ultra-Fast Analysis of Offshore Wind Turbine Support Structures using Impulse-Based Substructuring and Massively Parallel Processors", Proc. ISOPE 2015)

10. Single degree of freedom and effective damping: The focus on a single degree of freedom seems to be quite limiting. For idling loadcases (e.g. the storm condition

discussed in the manuscript), side by side motion of the turbine is expected to be important. Especially, if the waves are not aligned with the wind (although the authors assume the waves always to be aligned with the wind loads). It seems quite straightforward to include at least a second mode for the side-by-side motions. Why has this not been done?

11. Analysis of large turbines: There are indications that for 10MW+ wind turbines also a second tower deflection mode becomes excited. Again, this could be easily implemented. Please comment.

12. Differences in results: p16, l4: "The different results [...] must be because ..." - How can you be so sure? Replace by "seems likely because"?

13. Differences in the results: p16, l5f: "the assumption that Flex5 and QuLa can give the same tower deflections, which does not hold for all wind speeds" - Unclear what this is supposed to mean. Please explain and/or reformulate.

14. Ultimate load assessment: p17, l6: "In each interval the largest load is found and the average of these six loads calculated." - Is this an established procedure from a standard? Please give a reference and/or justification of the procedure.

15. Extreme wave analysis: p19, l21: "In other part of time series, though, the embedded stream function wave results ..." - It sounds as if the extreme wave events was embedded and assessed a number of times - how often?

16. Results in Figure 20: The shown example time series suggest that the response in QuLa is lower than in Flex5. However, the spectra shown suggest exactly the opposite. Please explain this apparent contradiction. Are the time series examples simply badly chosen, i.e., not representative? Or have the colors been mixed up, maybe?

17. Wave stretching: p21, l18f: "This difference could be improved by including Wheeler stretching in the model, which though would decrease the computational speed of the model" - If Wheeler stretching is so important for getting more accurate

results, could you explain a bit more why it would reduce the speed so much? Are the hydrodynamic loads not pre-computed as well (as there are no relative velocities used)? It should be simple to include stretching then, or where am I mistaken?

Minor comments:

- The citation style should be corrected. References should appear in brackets, not directly in the text. Example: "The 10MW DTU reference wind turbine (Back et al. 2013) ..." instead of "The 10MW DTU reference wind turbine Bak et al. (2013) ..."

- Abstract: What do you mean by "a load-based configuration"? Rephrase?

- Abstract: correct "both both"

- Abstract: "Some deviations for ..."

- Abstract: "The differences in ..."

- p1, l17: correct "manufactures"?

- p1, l22: "Furthermore" instead of "Further"?

- p2, l2: "in the time domain"?

- p2, l4: "drive-train, and controller"?

- p2, l14: Web address should be a footnote?

- p2, l23: "two different methods ... are discussed"?

- p2, l30: No commas in section titles

- p4, l16: Consider "Therefore it is the absolute" instead of "not the relative"

- Eq.6 and further: Notation "GM", "GK", "GD", "GF" is highly confusing. Variables should be given one letter only; additional subscripts can be used to differentiate. Change these names to "M_G", etc.?
[Figure]

Interactive
comment

- Eq.8 "GF" seems to be missing a hat.

- p9, l6: "van der Tempel's method"

- p11, l11: "foundation contributes significantly to the damping"

- p12, l2f: "Since fatigue loads are considered, the wind speed probability of occurrence ..."?

- p12, l6: "and the wind speeds are the same as for load case 1.2"?

- p12, l13: "applies to higher harmonics"?

- p12, l16f: ", which frequency or its multiples are equal..." - please rephrase.

- p12, l17: "However, due to ..."?

- p14, l12f: "exceedance probability curves of the moment peaks" - Consider rephrasing

- p15, l2:", here $N\_eq$ = ..."?

- p15, l9: correct "devitation"

- p15, l19: correct "The damping are ..."

- p15, l24: "Since QuLa only has one ..."?

- p15, l29: correct "dampng"

- p16, l1: "ratio of 0.93 relative to those of ..."?

- p17, l1: "it is important to note"?

- p17, l17: correct "ulimate"

- p19, l20: "In other parts of the time series..."?

- p21, l8: replace "most correct results" with "more correct results"
- p21, l9: correct "investiagte"

- p21, l10: rephrase "very close to be identical"

- p21, l14: "the values of QuLa vary"?

- Acknowledgments: missing? No project that financed this research?

---

## Author Comment (AC2) · 26 Jun 2017

**Answers to anonymous review**

The referee is thanked for the review. Answers and actions to all points are given below (blue text).

**General comments**

> It is important to highlight the issues and aspects
> - of the "quick" load calculation method that have been
> adressed here and have not been considered before.

- Which information is missing? The model is presented in the introduction and the following sections.

> - Does the overturning moment include both the
> fore-aft and side-side bending moments.

- No, only fore-aft. This will be added in beginning of section 2.2 "The structural model" where the shape function of the model is presented. Reason for fore-aft only is to have a simple and fast model.

> - please elaborate on how the aerodynamic damping
> has been precalculated.

- Which information is missing in the presentation of the methods to calculate the aerodynamic damping?

> - please consider the use of a list of
> symbols, since symbols used in the figures
> are not self explanatory.

- A list of symbols will be added.

- some information about the wind turbine would be useful since some parameters are mentioned many time such as, rated wind speed, rotational speed at rated etc.

  - Agree, this will be added, when the wind turbine is introduced on p.12, l.27.

- the damping behavior may change with the turbulence intensity, especially when the wind turbine is operating above rated wind speed, would the computed damping from the different methods change with turbulence intensity.

  - Considering the decay tests, it is not expected that the turbulent intensity would change the results much. As can be seen from figure 8, the damping ratio is very similar irrespectively of whether the wind speed is constant or turbulent, which is due to the fact, that the, pitch and rotor speed is kept constant. Considering the standard deviation method, the damping will change if the displacement changes, which is the case if the turbulent intensity is changed.

- the most critical state is usually around rated wind speed. In this case the wind speed fluctuate between below rated (where the pitch is not active) to above rated when the pitch is activated. How would this constant switching behavior affect the calculation of the damping ratio?

  - The damping is calculated for an average wind speed of 10 m/s and 12 m/s. For 12 m/s the pitch and rotor speed is kept constant. If the average wind speed of 10 m/s contained wind speeds above

rated, it was not found to cause any problems in decay tests. Perhaps the wind speed did not exceed rated in the 50 s the decay test ran for. But yes, the discussion is missing in the text. This will be added.

> *please be more specific on the loads that have been calculated with this method, bending moment in both directions?*

- No only fore-aft direction is considered. This will be added when the figures are presented.

> *directions?*
> *for the design of monopile/bucket, pile rotation could be important*

- Do you suggest that we should add a mode more? Rotations are included at 'lid' – refer to the spring in figure 2.

**Comments inside the paper**

All language and layout-issues will be corrected in the paper.

P5:

- No z is here in the horizontal direction, but I agree that this is confusing and will be change.
- The point force is function of $\eta_z$, and increases therefore as the wave becomes steeper. It is only significant for very large waves, and therefore it can be seen as a slamming force.

P7

1. No only fore-aft
2. Smaller will be replaced with lower.
3. Yes, gravity of both the RNA and the tower. This will be added in the text.

P8

2. Velocity produces damping through the GD term in eq (8). The pile displacement is just used for calibrating the linear damping coefficient. One could also have used standard deviation of tower top velocity. However, we do not expect difference to be large, at least not if the main motion occur at the natural frequency.
3. "Seen" will be changed to "visible".
4. Please see figure 6 and 7. For each decay test a run with and without an initial velocity is performed and afterwards subtracted.

P9

1. Text will be updated

2-3. Text will be updated. Blade rotational speed and rotor speed is the same.

4. The choice of a simple damping estimation is part of the models philosophy. We compare the model results to the full aero-elastic model FLEX5 in the paper to quantify how well the approximate steps applied work. This is done in figure 12-17.

P10

5. One explanation can be that the standard deviation puts more weight to the low-amplitude motion. In the decay tests, the damping seems to become smaller for low amplitude motion, see figure 7, lower plot, for t>30s. We will look into that during the revision and add a discussion to the paper.

P11

1. A situation with wake could also be considered, as this will change the turbulence intensity, and the aerodynamic forcing. The input to the model is therefore just changed.
2. Yes, soil damping is important to include. The soil damping is part of the viscous damping which represent structural damping, soil damping and hydrodynamic radiation, cf. page 6.

P13

The table text of table 2 will be updated.

As explained in the text on page 12 the "+" and "-" indicates, whether the peak frequency or multiples of the peak frequency are close to the natural frequency.

P17

It is an ultimate limit state, which is considered. This will be corrected in the text.

---

## Author Response (AR1)

**Answers to Michael Muskulus' review**

The referee is thanked for the review. Answers and actions to all points are given below (blue text).

1. New content: As this paper is part of a Special Issue of papers previously published with IOP (from The Science of Making Torque From Wind conference), I would have expected a footnote explaining this fact. As the authors are probably aware of, publication in Wind Energy Science journal is contingent on 40 percent new content compared to the previously published work. Can the authors explain a bit how they have updated the conference paper and what new results/content is included in the manuscript here?

Agree, this will we be added to the introduction. The introduction is extended and other models similar to the presented model is presented.

In the conference paper, the aerodynamic damping was only calculated by decay tests, and the method was not described in the paper. Further, in the conference paper, only load case 1.2 and load case 6.1 was considered. In the present paper, the comparison is more thoroughly and includes also load case 1.3.

2. Use of frequency domain: It is unclear if the phase information is retained or not. Are the coefficients \hat{alpha}\_j in Eq. 7 complex? The text ("can readily be transposed to the time domain by inverse FFT") suggests this is the case. If so, the solution is completely equivalent to a time domain integration. What is the reason for the use of the frequency domain then? The speedup due to the possibility of using FFT? Please comment and discuss in the text.

Yes, the phase information is retained, and yes the reason for the use of the frequency domain is due to the speed-up. It is now added in the text that "\hat{\alpha}\_j is a complex number, \omega\_j is the smallest angular frequency in the time series and c.c. is the complex conjugate. The phase information of \$\alpha\$ is retained in (7)." After (8) is now written that "By solving the equation in frequency domain the solution \hat{\alpha} can then be solved at once for all time steps".

3. Aerodynamic damping: p7, l21f: "... it is necessary to simplify the aerodynamic and add the damping [...] as a viscous linear damping force ..." - This seems a bit too suggestive. Why is it "necessary" to model the damping with a linear viscous damper? (In fact, the aerodynamic damping force is definitely not linear) Agree, it is not necessary, but the aerodynamic damping can only be added as a linear viscous damper. This is rephrased in the paper:

"....the aerodynamic damping can only be added as a viscous linear damping force, where the damping coefficient is a function of the mean wind speed."

4. Calculation of standard deviation: Eq. 15 seems to have some issues. First, why the factor of 1/2? What is the summation over? How is the displacement \hat{u} determined from the previous \hat{\alpha} - is it the same? And should it not be an absolute square of the (complex?) displacements? The equation will be changed to time domain as this is easier to interpret: Sigma=sqrt(mean[(u-mean(u))^2])

5. Determination of damping ratio: p9, l9f: "The damping [...] is found by keeping the pitch and rotor speed constant, since it is a very simple method which can be reused several times" - Unclear what the latter part of this sentence refers to. Please explain.

Bad phrasing. The text is irrelevant here and is deleted.

6. Determination of damping ratio: p10, l1f: "The logarithmic decrement is the average of the four peaks ..." - Imprecise formulation. Did you mean: "The logarithmic decrement has been estimated for the four peaks ... and then averaged"?

Correct, the text is updated: "The logarithmic decrement has been estimated for the four peaks, both positive and negative, following the largest peak and then averaged".

7. Discussion of damping: In general, the term "damping" is used somewhat ambiguously. Without further explanation, I would assume it stands for a damping force, but the authors seem to use it mostly for the "damping ratio". Please consider a more precise use. Also, p13, l1: It seems unusual to report the logarithmic decrement in percent - percent of what? This is normally used for damping ratios only (percent of critical damping), and therefore misleading here

The text is updated, and it is specified whether it is the damping force, damping ratio and so on, which is considered. The logarithmic damping on page 13 is changed to logarithmic decrement.

8. ULS wave loads: p12, l25: How were the values giving the largest wave loads in the interval from Eq. 20 determined? Values are given in line 25, but how were they found?

For load case 6.1 the calculations were made for 6 different wave periods from 11.1\sqrt{Hs/g} to 14.3\sqrt{Hs/g}. It was found that the largest load was found for the smallest wave period. This is explained in the text:

"In the present analysis six wave periods from 11.1\sqrt{Hs/g} to 14.3\sqrt{Hs/g} was considered for load case 6.1. It was found that for the present structure the largest load occurred for the smallest wave period, T=11.1\sqrt{H\_s/g}=10.87\$s. In load case 1.3 the same ratio, T=11.1\sqrt{H\_s/g}\$ is also used."

9. Focus on speed: The main motivation for the method seems to be that it results in much faster load simulations. However, alternatives exist that should at least be discussed, e.g. the convolution-based approach in the time domain (Schafhirt et al.: "Ultra-Fast Analysis of Offshore Wind Turbine Support Structures using Impulse-Based Substructuring and Massively Parallel Processors", Proc. ISOPE 2015) Agree, the paper is now mentioned in the introduction. However, this method still require that the wind turbine manufacture share information about their wind turbines, which is difficult to ensure.

10. Single degree of freedom and effective damping: The focus on a single degree of freedom seems to be quite limiting. For idling loadcases (e.g. the storm discussed in the manuscript), side by side motion of the turbine is expected to be important. Especially, if the waves are not aligned with the wind (although the authors assume the waves always to be aligned with the wind loads). It seems quite straightforward to include at least a second mode for the side-by-side motions. Why has this not been done? It is straight-forward to add a side-side degree of freedom. This could be done for extended analysis. We have here focused on simplest possible and fastest approach.

11. Analysis of large turbines: There are indications that for 10MW+ wind turbines also a second tower deflection mode becomes excited. Again, this could be easily implemented. Please comment. We agree this can be addded easily. This forexample done in Smilden et al. (2016) "Reduced Order Model for Control Applications in Off shore Wind Turbines". A reference to this paper will be added in the text. It will essentially double the complexity of the model. So far our approach has been to make the model simplest possible and quantify how accurate results one gets.

12. Differences in results: p16, l4: "The different results [...] must be because ..." - How can you be so sure? Replace by "seems likely because"? This will be corrected.

13. Differences in the results: p16, I5f: "the assumption that Flex5 and QuLa can give the same tower deflections, which does not hold for all wind speeds" - Unclear what this is supposed to mean. Please explain and/or reformulate. '

This is rephrased: "...are based on the assumption that the tower deflection is the same for Flex5 and QuLA, which is not correct for all wind speeds."

14. Ultimate load assessment: p17, I6: "In each interval the largest load is found and the average of these six loads calculated." - Is this an established procedure from a standard? Please give a reference and/or justification of the procedure.

The following is now added to the text:

"This approach is consistent with the IEC 61400-3 code, clause 7.5.1 for load case 1.3. For load case 6.1a, clause 7.5.1 states that six 1 hour realizations should be considered, unless it can be demonstrated that the extreme response is not affected by application of shorter realizations. Constrained wave methods is mentioned as one way of enabling shorter realizations. This approach has been adopted for the present study. For some realizations, we found that the largest loads occurred at events outside of the embedded constrained wave. This is further discussed in section 4.2.2."

15. Extreme wave analysis: p19, l21: "In other part of time series, though, the embedded stream function wave results ..." - It sounds as if the extreme wave events was embedded and assessed a number of times - how often?

A one hour times series is considered, and divided into 6x600 s intervals. The largest wave in each interval is replaced with a stream function wave.

16. Results in Figure 20: The shown example time series suggest that the response in QuLa is lower than in Flex5. However, the spectra shown suggest exactly the opposite. Please explain this apparent contradiction. Are the time series examples simply badly chosen, i.e., not representative? Or have the colors been mixed up, maybe?

It is correct that for this part of the time series, the response in QuLA is smallest. However, this does account for the whole time series, and the spectra are based on the whole time series. This is explained in the text:

"In the tower, the energy of the force and moment is located around the first natural frequency and it is clear that QuLA contains most energy at this frequency. This is opposite to the time series, which indicate that the response of Flex5 contains most energy. However, this does not account for the whole time series, which the spectra are based."

17. Wave stretching: p21, l18f: "This difference could be improved by including Wheeler stretching in the model, which though would decrease the computational speed of the model" - If Wheeler stretching is so important for getting more accurate results, could you explain a bit more why it would reduce the speed so much? Are the hydrodynamic loads not pre-computed as well (as there are no relative velocities used)? It should be simple to include stretching then, or where am I mistaken?

The wave kinematics and forces are computed 'on the fly' by FFTs. This allows the subsequent force computation to be fast if it is done on the same fixed z-levels. Hence consistent linear integration up to z=0 can be done fast. When you include Wheeler stretching, the grid spacing of the z-vector changes. Therefore, you have to make the integration of the forces for each time step individually. This slows the process down.

We agree that stretching could be included as part of the pre-computation of loads, and afterwards the wave kinematics could be interpolated to fixed z-values. Then one would have to store the wave kinematics up along the pile to allow for later use in Morison with varying design diameters. We chose the simpler approach of no stretching.

Minor comments The minor comments will be corrected and acknowledgement included.

**Answers to anonymous review**

The referee is thanked for the review. Answers and actions to all points are given below (blue text).

**General comments**

It is inportant to highlight the issues and aspects - of the "quick" load calculation method that have been adressed here and have not been considered before. me transmiss include both the

 Which information is missing? The model is presented in the introduction and the following sections.

Does the overturning moment include both the fore-aft and side - side bending moments.

No, only fore-aft. This is added in beginning of section 2.2 "The structural model" where the shape function of the model is presented.
*"Only one shape function in the fore-aft direction is considered in QuLA."* Reason for fore-aft only is to have a simple and fast model.

- please elaborate on how the accodynamic damping has been precalculated.

Which information is missing in the presentation of the methods to calculate the aerodynamic damping?

- please consider the use of a list of symbolo, since symbols used in the figures are not self explanatory.

- A list of symbols will be added.

- some information about the wind tendine would be useful since some parameters are inentioned many time such as , rated wind speed, rotational speed at rated etc.

Agree, this will be added, when the wind turbine is introduced on p.12, l.27.

the damping behavior may change the turbulence intensity, especially when the wind turbine is operating above rated wind speed, would the computed damping from the different methods change with turbulence intensity.

Considering the decay tests, it is not expected that the turbulent intensity would change the results much. As can be seen from figure 8, the damping ratio is very similar irrespectively of whether the wind speed is constant or turbulent, which is due to the fact, that the, pitch and rotor speed is kept constant. Considering the standard deviation method, the damping will change if the displacement changes, which is the case if the turbulent intensity is changed.

most critical state is usually around rated wind speed. In this case the wind speed Auctuate between below nated (where the pitch is not achie) to above rated where the pitch is activated. How would this constant switching behavior affect the calculation of the damping ratio?

- The damping is calculated for an average wind speed of 10 m/s and 12 m/s. For 12 m/s the pitch and rotor speed is kept constant. The average wind speed of 10 m/s did not contain wind speeds above rated, and was therefor not found to cause any problems in decay tests.

please be more specific on the loads that have been calculated with this method, bending moment in both directions ?

- No only fore-aft direction is considered. This is explained in the beginning of section 4.1: *"Only the deflection in the fore-aft direction is calculated in QuLA and therefore only the forces and moments in the fore-aft direction are considered in the analysis."*
- tor the design of monopile/bucket, pile rotation could be important
- Do you suggest that we should add a mode more? Rotations are included at 'lid' refer to the spring in figure 2.

**Comments inside the paper**

All language and layout-issues will be corrected in the paper.

P5:

- No z is here in the horizontal direction, but I agree that this is confusing and will be change.
- The point force is function of \eta\_z, and increases therefore as the wave becomes steeper. It is only significant for very large waves, and therefore it can be seen as a slamming force.

Ρ7

- 1. No only fore-aft
- 2. Smaller will be replaced with lower.
- 3. Yes, gravity of both the RNA and the tower. This will be added in the text.

**P8**

- 2. Velocity produces damping through the GD term in eq (8). The pile displacement is just used for calibrating the linear damping coefficient. One could also have used standard deviation of tower top velocity. However, we do not expect difference to be large, at least not if the main motion occur at the natural frequency.
- 3. "Seen" will be changed to "visible".
- 4. Please see figure 6 and 7. For each decay test a run with and without an initial velocity is performed and afterwards subtracted.

Ρ9

1. Text is updated:

"The logarithmic decrement is calculated from the difference between the two simulations with and without a starting velocity."

2-3. Text is updated. Blade rotational speed and rotor speed is the same.

4. The choice of a simple damping estimation is part of the models philosophy. We compare the model results to the full aero-elastic model FLEX5 in the paper to quantify how well the approximate steps applied work. This is done in figure 12-17.

**P10**

5. This might be because the standard deviation puts more weight to the low-amplitude motion. In the decay tests, the damping seems to become smaller for low amplitude motion, see \ref{fig:DecayT} lower plot, for \$t>30\,\$s. The reason there is a large increase in the damping ratio based on the tower top displacements around \$W=17\$ m/s can be because this method assumes that the tower deflection is the same for Flex5 and QuLA. This is not correct as explained in section \ref{sec:eigen}. As the wind speed increases the tower has a larger deflection and the contribution from the gravity therefore larger. This contribution is not included when QuLA calculates the deflection.

**P11**

- 1. A situation with wake could also be considered, as this will change the turbulence intensity, and the aerodynamic forcing. The input to the model is therefore just changed.
- 2. Yes, soil damping is important to include. The soil damping is part of the viscous damping which represent structural damping, soil damping and hydrodynamic radiation, cf. page 6.

**P13**

The table text of table 2 is updated.

As explained in the text on page 12 the "+" and "-" indicates, whether the peak frequency or multiples of the peak frequency are close to the natural frequency.

**P17**

It is an ultimate limit state, which is considered. This will be corrected in the text.

---

## Referee Report (RR1)

[referee-annotated manuscript omitted]

---

## Author Response (AR2)

**Answers to Michael Muskulus' review**

The referee is thanked for the review. Answers and actions to all points are given below (blue text).

1.

The frequency domain method is used with phases included here, which is somewhat unusual - but of course nothing wrong with this. It just means that the method is equivalent to time domain calculations. Computationally, there is not much to be gained here, since for this decoupled analysis the use of an impulse-response-function (as by Schafhirt et al.) can be evaluated with the FFT as well. The observation (p2, l1) that the use of impulse-response-functions needs the manufacturer to share confidential information is not accurate. The only information that is needed is a number of impulse-response-functions, which is similar to the situation in QuLA, where also some information about the support structure, e.g., a mode shape is needed. This should be corrected in the text. I would also suggest to mention already in the abstract that the frequency domain "with phases" is used, for the benefit of readers.

We apologies for the misunderstanding regarding the response functions. The line *"However, this method still require that the wind turbine manufacture share information about their wind turbines, which is difficult to ensure."* is deleted. We have added to line XX that the work of Shafhirt et al concerns rotor loads:

"Recently Schafhirt et al. (2015) combined a sub structuring technique, which is based on the principle of superposition of impulse responses, with the power of modern general purpose graphics processing units to compute the response of an offshore wind turbine subject to rotor loads."

It is added in the abstract that phases are included in frequency-domain:

"The dynamic structural response is represented by the first global fore-aft mode only and is computed in the frequency domain with phases using the equation of motion."

**2.**

p6, l14: I assume that the reason why the wave kinematics are transformed to the time domain are the nonlinearities in the hydrodynamic force calculation? (Otherwise one could solve the equation of motion directly in the frequency domain) This could be mentioned here explicitly.

The sentence on page 6 line 15-16 is changed:

"The linear irregular wave kinematics and loads are calculated in the frequency domain and afterwards transformed to the time domain using inverse Fast Fourier transformation in order to include the nonlinear terms in the hydrodynamic force."

3.

Small textual corrections

p11, l8: "is calculated alternatively in Flex5"?
p13, l13: "see Fig. 7, lower plot, "?
p16, l2: "rated rotor speed"?
p18, l12: "decay tests results in the best agreement"?

p21, l10: "to the moment"?p21, l17f: "are part of the exceedance"?p22, l9: "which the spectra are based on."?

This is corrected.

**Answers to anonymous review**

The referee is thanked for the review. Answers and actions to all points are given below (blue text).

**Page 4**

 $\begin{array}{c} z & \text{Vertical coordinate} \\ \lambda & \underline{\text{Constant i eq. (1)}} \\ \alpha & \text{Generalized coordinate} \\ \end{array} \\ \begin{array}{c} \text{Shear segment} \\ \text{for the power law} \end{array} \\ \begin{array}{c} ? \\ \text{Changed. Below equation (1) the following is now added:} \end{array} \\ \end{array}$

"Usually the shear exponent is designated as \alpha, but since \alpha in this paper represent the generalized coordinate, the shear exponent is instead designated as \lambda.

Please add the units, S, KN, N.

Units are added to symbol-list

**Page 18**

I what does increase down through means

The sentence is changed to " increases from the top of the monopile and down to the bottom."

(2) the underestimation of the load by Quild can be compensated somehow? To make sure the design loads are appropriate.

Yes, it could be compensated by including an extra mode in QuLA. This is explained in line 25-27 on page 28:

"The main part of the modal energy of the second natural frequency is distributed in the Mono Bucket, which explains why the difference between the two models at the second natural frequency is largest at the sea bed and why the ratio of the equivalent forces in figures  $\figures \FLS12Decay$ - $\fig:FLS12Standard$  decreases throughout the Mono Bucket."

The following lines is added:

"This difference could be reduced by including a second degree of freedom in QuLA, as was done by Smilden etal (2016). However, this will also double the complexity of the model, and focus has been to develop a very simple and fast model."